# Assessing soil redistribution of forest and cropland sites in wet tropical Africa using [239+240]Pu fallout radionuclides

Florian Wilken[1,2], Peter Fiener[2], Michael Ketterer[3], Katrin Meusburger[4], Daniel Iragi Muhindo[5], Kristof van Oost[6], Sebastian Doetterl[1]

[1]Department of Environmental Systems Science, Eidgenössische Technische Hochschule Zürich, Zürich, Switzerland
[2]Institute for Geography, Universität Augsburg, Augsburg, Germany
[3]Chemistry and Biochemistry, Northern Arizona University, Flagstaff, USA
[4]Swiss Federal Institute for Forest, Snow and Landscape Research, Birmensdorf, Switzerland
[5]Faculty of Agronomy, Université Catholique de Bukavu, Bukavu, DR Congo
[6]Earth and Life Institute, Université Catholique de Louvain, Louvain-la-Neuve, Belgium

*Correspondence to:* Florian Wilken (florian.wilken@usys.ethz.ch)

**Abstract**

Due to the rapidly growing population in tropical Africa, a substantial rise in food demand is predicted in upcoming decades, which will result in higher pressure on soil resources. However, there is limited knowledge on soil redistribution dynamics following land conversion into arable land in tropical Africa that is partly caused by infrastructure limitations for long-term landscape scale monitoring. In this study, fallout radionuclides [239+240]Pu are used to assess soil redistribution along topographic gradients at two cropland sites and at three nearby pristine forest sites located in the DR Congo, Uganda and Rwanda. In the study area, a [239+240]Pu baseline inventory is found that is higher than typically expected for tropical regions (mean forest inventory 41 Bq m$^{-2}$). Pristine forests show no indication for soil redistribution based on [239+240]Pu along topographical gradients. In contrast, soil erosion and sedimentation on cropland reached up to 37 cm (81 Mg ha$^{-1}$ yr$^{-1}$) and 40 cm (87 Mg ha$^{-1}$ yr$^{-1}$) within the last 55 years, respectively. Cropland sites show high intra-slope variability with locations showing severe soil erosion located in direct proximity to sedimentation sites. This study shows the applicability of a valuable method to assess tropical soil redistribution and provides insight into soil degradation rates and patterns in one of the most socio-economically and ecologically vulnerable regions of the world.

## 1. Introduction

Soil erosion is considered to be the major threat to global soil resources and substantially contributes to crop yield reduction (Amundson et al., 2015; Montanarella et al., 2016; Govers et al., 2017), which challenges food security in regions facing population growth beyond sustainable limits in the 21st century. In particular, the White Nile-Congo rift (NiCo) region faces a strong impact of soil redistribution (Lewis and Nyamulinda, 1996; FAO and ITPS, 2015; Montanarella et al., 2016) and corresponding sediment delivery (Vanmaercke et al., 2014) due to steep terrain, high rainfall erosivity with a strong intra-annual seasonality (Fick and Hijmans, 2017) that causes sparse vegetation cover of soil at the end of the dry seasons but also throughout the cultivation period (Lewis and Nyamulinda, 1996; Fick and Hijmans, 2017) due to non-mechanised farming. The region is also predicted to undergo substantial climate change, which might further increase soil erosion (Borrelli et al., 2020). The loss of soil resources and crop yield decline in the NiCo region happens in parallel to a rapid population growth (population of Rwanda, Uganda and Democratic Republic of Congo 2020: 150 millions – predicted 2100: 430 millions; WPR, 2020), which drives rising food demands that are expected to triple for the entire Sub-Saharan Africa between 2010 and 2050 (van Ittersum et al., 2016). The associated pressure on land resources leads to various problems that will have dramatic ecological and social impacts (food insecurity, political unrest, migration) in the NiCo region (Chamberlin et al., 2014; FAO and ITPS, 2015). Under current practices, an increasing demand in food is typically compensated through deforestation to assess new soil resources, which are often located in areas with steep slopes (Govers et al., 2017). This causes a loss of highly valuable forest ecosystem services (e.g. carbon storage, biodiversity, imbalance of the hydrological cycle) and the onset of potential soil erosion at previously undisturbed sites (Nyssen et al., 2004) and often unsustainable use of soil resources (Wynants et al., 2019). In steep cropland sites of the NiCo region, it is frequently observed that the entire deep tropical saprolite body is removed and the bedrock is exposed at the surface, which means a quasi-permanent loss of cropland and the option to reforest areas for decadal to centennial timescales (Evans et al., 2020). A pressing need persists to predict future dynamics and timescales of cropland degradation in order to understand the pace of a rising food shortage and to develop adapted agricultural management strategies at suitable locations. Smart intensification of existing cropland soils due to adapted agricultural practices in suitable locations and the conservation/restoration of soils prone to erosion (e.g. reforestation or grassland use) has been discussed as a key management strategy to combat degradation (Govers et al., 2017). To develop a smart intensification plan detailed information on soil degradation dynamics of individual regions under specific conditions is essential (e.g. land use, topography, soil type, rainfall characteristics). Soil erosion plot experiments were carried out in tropical Africa (Lewis and Nyamulinda, 1996; Xiong et al., 2019) to understand the rates of soil loss. However, plot experiments are limited to soil erosion processes, while soil redistribution dynamics (interaction of erosion and sedimentation) remain unexplained, although they are crucial to understand soil degradation rates on a landscape-scale. However, catchment monitoring that provides insight into spatially distributed soil redistribution dynamics necessitates a sufficiently long time series (years to decades) to integrate a statistically representative variety of erosive rainfall events impacting different land cover conditions (Fiener et al., 2019). Particularly in regions of limited infrastructure, long-term catchment monitoring projects

are challenging and typically rare. This problem can be overcome by the analysis of fallout radionuclides from nuclear weapon tests (i.e. $^{137}$Cs, $^{239+240}$Pu) as soil redistribution tracers (Meusburger et al., 2016; Alewell et al., 2017; Evrard et al., 2020), which have the major advantage to provide insight into spatial patterns of both soil erosion and sedimentation processes integrated over the period since 1963-1964 (Test Ban Treaty that caused a rapid decrease of atmospheric bomb tests; Wallbrink and Murray, 1993; Evrard et al., 2020). The most widely used fallout radioisotope in soil redistribution studies is $^{137}$Cs (e.g. Porto and Walling, 2012; Chartin et al., 2013; Evrard et al., 2020), which has a rather short half-life of about 30 yrs. Hence, decay has already led to a pronounced reduction (73% in 2020) of the $^{137}$Cs activity until today (Alewell et al., 2017). In tropical and equatorial regions, this is a critical limitation of using $^{137}$Cs for soil redistribution analysis due to much lower fallout compared to the mid latitudes of the northern hemisphere (Hardy et al., 1973; Evrard et al., 2020). Furthermore, extreme soil erosion rates in the tropics (Lewis and Nyamulinda, 1996; Angima et al., 2003; Nyesheja et al., 2019; Xiong et al., 2019) are additionally driving a depletion of the $^{137}$Cs inventories. To overcome these analytical difficulties, the fallout radionuclides $^{239}$Pu and $^{240}$Pu have been discussed and tested as an alternative radioisotopic tracer to $^{137}$Cs for soil redistribution studies over the past decade. The major advantage of both isotopes is their long half-life ($^{239}$Pu = 24,110 yrs, $^{240}$Pu = 6,561 yrs). Furthermore, $^{239}$Pu and $^{240}$Pu show a very limited plant uptake (Akleyev et al., 2000) and preferentially form associations with soil iron oxides (Ryan et al., 1998; Lal et al., 2013), which potentially makes the isotopes very suitable tracers for tropical environments dominated by Ferralsols. Hence, the $^{239+240}$Pu activity in tropical soils might be high enough to successfully carry out soil redistribution studies and provide important insights on soil redistribution dynamics in tropical Africa. Few fallout radionuclide based soil redistribution studies have been carried out in the tropics (Evrard et al., 2020) and, to the best of our best knowledge, non was conducted in the wet tropics of Africa (Af, Am climate; Kottek et al., 2006).

The aims of the study are (i) to test the suitability of fallout radionuclides $^{239}$Pu and $^{240}$Pu as tracers of soil redistribution in the wet tropics of Africa, and (ii) to analyse soil redistribution dynamics after conversion from forest into cropland for selected sites within the East African NiCo region.

## 2. Methods

### 2.1 Study sites and sampling design

The NiCo region is located in the headwater catchments of the White Nile (Lake Edward) and the Congo River (Lake Kivu) that are part of the East African Rift Valley system (Fig. 1). The region faces rapid population growth creating substantial pressure on land resources and initiating conversion of forest into cropland. Soil degradation by water erosion is a recognised problem in the region (Lewis and Nyamulinda, 1996; Montanarella et al., 2016) indicated via frequent soil erosion events resulting in ephemeral rills and gullies as well as permanent deep gully systems. The region is characterised by steep terrain (Fig. 1; Tab. 1) and tropical climate with a mean annual air temperature between 16.7 and 19.3°C and an annual rainfall ranging between 1300 and 1900 mm (time period 1970-2000; Fick and Hijmans, 2017). The seasonal rainfall distribution is subdivided into two cycles of wet and dry seasons (Fick and Hijmans, 2017). The rainfall erosivity in the region is high due to frequently

occurring storm events with large rainfall amounts linked to high rainfall intensities during the rain seasons (on average 20 erosive rainfall events per rain season; events exceeding 10 mm h$^{-1}$ of rainfall per 30 min interval)(Doetterl et al., 2021b). Soils in the region are deeply weathered Ferralsols (> 6 m; WRB, 2006; Doetterl et al., 2021a) developed from three geochemically distinct parent materials (DR Congo: mafic magmatic rocks; Uganda: felsic magmatic rocks; Rwanda: sedimentary rock of mixed geochemical composition). Soils across the study area are typically classified as clay loam while at the Ugandan forest and cropland study sites a lower clay and higher sand content is found (Doetterl et al., 2021a).

The forest sites in the study area are primary tropical mountain forests (for detailed information see Doetterl et al., 2021a). Farming is documented since the 1950s for the site in DR Congo, while conversion into cropland at the Ugandan site took place during the 1970s (personal communication with local villagers). The cropland sites represent the typical smallholder farming found in the region, which is based on small non-terraced fields with non-mechanised tillage practices. Due to the small fields (mean field size = 450 m$^2$) and an individual and dynamic field management, a high patchiness of soil cover conditions is present and can alternate between bare soil and fully grown vegetation cover in direct proximity to each other (Fig. A1).

In 2018, a soil sampling campaign was carried out in three pristine forests and two cropland sites in the NiCo region in order to collect soil samples for a soil redistribution assessment based on the fallout radionuclides $^{239}$Pu and $^{240}$Pu. As part of this campaign, a total of 347 samples were taken. Soil sampling was carried out using a manual closed tube soil corer (VSI soil core sampler, Vienna-Scientific, Austria) with a diameter of 6.8 cm and a length of 120 cm.

Sampling sites in forests were located within the Kahuzi Biega (DR Congo), Kibale (Uganda) and Nyungwe Forest (Rwanda) National Parks (Fig. 1). There, the sampling scheme is aligned to a toposequence approach that covers three different landscape positions situated along a catena: plateau, slope and foot-slope. Sampling was carried out at 19 locations within each catena and covers different soil layers (L-horizon, O-horizon and mineral soil; see Tab. 2). Discrete sampling at plateau locations took place in order to quantify spatial variation in reference fallout inventories. To average out the typical variability of fallout radionuclide inventories, composite samples were taken at the slope and foot-slope positions. In addition to mineral soil layers, two organic layers at two different levels of decomposition (L and O horizons) were collected over an area of 20 cm x 20 cm at each sampling site. About 40% of the total forest samples were taken at plateau positions (high proportion due to non-composite sampling), while 30% of the samples were taken at slope and foot-slope locations, respectively. At foot-slope locations soils were additionally sampled from 60 cm to 120 cm soil depth to cover colluvial sites with $^{239+240}$Pu activity in deeper soil layers.

Sampling sites in cropland were placed along two catenae within DR Congo and in Uganda, covering 51 individual locations at each study site (Tab. 2). The majority (~50%) of sampling sites are distributed along slope positions (12-13° steepness in both cropland sites) while 25% of sites are located at foot-slope and plateau sites. To understand the depth distribution of $^{239+240}$Pu and the variability of radionuclide inventories under stable geomorphic conditions, three depth increments of 20 cm thickness were taken to a total soil depth of 60 cm at plateau sites. In DR Congo, a cropland plateau site (converted to grassland

approximately in 2005) located about 8 km apart from the study site was sampled, while in Uganda, flat plateau sites under arable use in direct proximity to the study site were sampled. At slope locations, a single soil increment down to 60 cm was taken. At the foot-slope locations, to cover potential sedimentation, an additional increment from 60 to 100 cm was sampled to assure full cover of the radionuclide inventory.

## 2.3 $^{239+240}$Pu measurements

An assessment of fallout radionuclides $^{239+240}$Pu inventories was used to estimate effective soil redistribution since the 1960s along the investigated geomorphic transects. Plutonium isotopes measurements were conducted following Calitri et al. (2019) and Ketterer et al. (2004). The chemical preparation consisted of the following sequence:

1. Soil material was milled and subsequently dry-ashed for at least eight hours at 600°C to remove organic matter. An aliquot of the dry-ashed material of up to 50 grams was weighed into a 250 mL polypropylene bottle. Samples were tested for excessive reaction of carbonates by addition of 5 mL of 8 M $HNO_3$.

2. Samples were spiked using 7 picograms (~ 1 mBq) of a $^{242}$Pu tracer (NIST 4334g), in the form of a solution in 4 M aqueous $HNO_3$.

3. 125 mL of 8 M aqueous $HNO_3$ were added, with caution being exercised to add acid slowly when carbonates were present.

4. The sample vessels were capped and heated at 80°C overnight (~ 16 hours) with occasional mixing.

5. Following heating, the sample leach solutions were recovered by filtration with 0.45 micron cellulose nitrate membranes.

6. The plutonium was converted to the +4 oxidation state via addition of 0.5 g $FeSO_4*7H_2O$ dissolved in 2 mL water, followed by 2 grams of $NaNO_2$ dissolved in 5 mL water. Thereafter, the solutions were heated uncapped in a 90° C convection oven for 1.5 hours to release evolved $NO_2$ (g) and allow for conversion of the Pu into Pu(IV).

7. 300 milligrams of Pu-selective resin TEVA (EIChrom, Lisle, IL, USA) was added to the sample solution; the mixtures were agitated over a 4-hour timeframe to allow for the resin to uptake the Pu(IV).

8. The TEVA resin was collected on a 20 mL polyethylene column equipped with a glass wool plug; the pass-through solution was drained and discarded. The columns were rinsed with the following sequence: i) 50 mL of 2 M aqueous $HNO_3$; ii) 20 mL of 9 M aqueous HCl; and iii) 10 mL of 2 M aqueous $HNO_3$. The rinse sequence removes matrix elements, uranium and thorium.

9. Plutonium was eluted from the columns using the following sequence: i) 0.5 mL water; ii) 0.5 mL of 0.05 M aqueous ammonium oxalate; and iii) 0.5 mL water, all of which were collected together for analysis directly after elution.

In the preparations, quality control samples were included to assess the results; these consisted of blanks (35 g powdered sandstone devoid of detectable Pu). Blanks consisting of 35 g sandstone spiked with small quantities (50-100 milligrams) of IAEA 384 (Fangataufu Sediment) were also prepared. Eleven measurements of IAEA 384 resulted in an average of 102 Bq kg$^{-1}$ $^{239+240}$Pu, and a standard deviation of 10 Bq kg$^{-1}$; these results compare well to the reference value of 107 Bq kg$^{-1}$ $^{239+240}$Pu. The blanks were used to determine a detection limit of 0.01 Bq kg$^{-1}$ $^{239+240}$Pu.

Sample Pu fractions were measured with a Thermo X Series II quadrupole ICP-MS (Bremen, Germany) equipped with an APEX HF high-efficiency sample introduction system (ESI Scientific, Omaha, NE, USA). The APEX is equipped with a self-aspirating concentric fluorinated ethylene-propylene nebulizer operating at an uptake rate of ~ 0.15 mL per minute. The instrument is located at Northern Arizona University; the laboratory is licensed with the State of Arizona for handling $^{242}$Pu spike solutions. The intensities of $^{235}$U, $^{239}$Pu, $^{240}$Pu and $^{242}$Pu were recorded using a peak-jump algorithm (10 ms dwell time,

1000 sweeps/integration, three integrations per sample). The $^{235}$U isotope was measured as a proxy for $^{238}$U, the latter whose intensity occasionally exceeded the linear range of the ICPMS's pulse-counting detector; this was done in order to assess the potential interference of $^{238}$U$^1$H$^+$ on $^{239}$Pu. As many samples exhibited relatively high levels of $^{238}$U$^1$H$^+$, generating a potential "false-positive" detection of $^{239}$Pu, it was determined to be advantageous to measure Pu using the $^{240}$Pu isotope, which is unaffected by uranium hydride species. The measured atom ratio $^{240}$Pu/$^{242}$Pu was converted into a mass of $^{240}$Pu detected, using

the known mass of $^{242}$Pu; the $^{240}$Pu activity was calculated, and converted into the corresponding $^{239+240}$Pu activity, based on the known $^{240}$Pu/$^{239}$Pu atom and activity ratios in stratospheric fallout Pu (Kelley et al., 1999). The method resulted in a detection limit of 0.01 Bq kg$^{-1}$ $^{239+240}$Pu, which equates to a soil inventory of ca. 5-8 Bq m$^{-2}$ $^{239+240}$Pu for a bulk density of ~0.8 to ~1.3 Mg m$^{-3}$.

**2.4 Cropland soil redistribution calculation**

To derive topographic change and corresponding soil redistribution using $^{239+240}$Pu inventories, a mass balance model (Zhang et al., 2019) was applied integrating soil erosion and sedimentation over the period 1964 to 2018. Soil erosion and sedimentation processes were implemented (R-Core-Team, 2019) individually to account for differences of both processes. Due to topsoil loss by soil erosion, former soil from deeper layers with negligible low $^{239+240}$Pu activity gets increasingly incorporated into the plough layer. This exponential decay of the $^{239+240}$Pu inventory is mainly controlled by the soil erosion

magnitude and frequency and plough depth. These processes are addressed by the mass balance model that simulates the $^{239+240}$Pu inventory reduction on a year by year basis over the simulation period:

$$A_{yr} = \int_{1964}^{yr} A_{yr-1963} \left(1 - \frac{R_{yr}}{d}\right)$$

where $yr$ is the simulation year, $A_{yr}$ is the $^{239+240}$Pu inventory at the specific simulation year in Bq m$^{-2}$, $A_{yr-1963}$ is the annually updated $^{239+240}$Pu inventory in Bq m$^{-2}$, $d$ is the average plough depth in m, $R_{yr}$ is soil erosion of simulation year in m. Based

on these results, an individual logarithmic function between $A$ and $R$ for each study site was fitted that can be used to derive the amount of soil loss in cm (55 yrs.)$^{-1}$.

Sedimentation is represented as a linear increase of the inventory:

$$R = -d \left(1 - \frac{A}{A_{ref}}\right)$$

where $R$ is the soil redistribution rate in m (55 yrs)$^{-1}$, $A_{ref}$ is the $^{239+240}$Pu reference inventory and $A$ is the $^{239+240}$Pu activity difference (reference vs. local activity) in m x m (55 yrs)$^{-1}$ and Bq kg$^{-1}$ x Bq kg$^{-1}$ (55 yrs)$^{-1}$ over the simulation period (more details on the $A_{ref}$ implementation are given in section 2.5).

Finally, the $^{239+240}$Pu activity (Bq m$^{-2}$) was converted from topographic change (in m) into soil redistribution rates $Y$ in Mg ha$^{-1}$ (55 yrs)$^{-1}$ as follows (Walling et al., 2011):

$$Y = 10 \, d \, B \left(1 - \frac{R}{d}\right)$$

where $B$ is the bulk density in kg m$^{-3}$. $B$ is measured for each mineral and O-horizon sample, while $B$ for the L layer was taken from the literature, i.e. 80 kg m$^{-3}$ (Wilcke et al., 2002).

**2.5 Cropland scenario assessment using a mass balance model**

The mass balance soil mixing model was used to assess different scenario assumptions and their sensitivity. First, different $^{239+240}$Pu reference inventories were determined in two ways: (i) the mean $^{239+240}$Pu inventory of all forest sites of a specific region (Ref$_{for}$; i.e. mean inventory of Kahuzi Biega forest for the DR Congo cropland sites and Kibale forest for the Ugandan cropland sites) and (ii) the mean $^{239+240}$Pu inventory of the cropland plateau sites of the specific region (Ref$_{plt}$). These land use specific reference scenarios are supposed to address potential differences between Ref$_{for}$ and Ref$_{plt}$ to understand uncertainties associated with the reference determination. Second, the sensitivity of the ploughing and corresponding mixing depth is assessed using a 20±5 cm ploughing depth deviation as the exact plough depth over the integration period cannot be accurately derived. Third, to address potential interannual variability of water erosion, a scenario with five extreme years producing the same total soil erosion as a 55 years continuous soil erosion rate was compared against the results of the first scenario.

**3. Results**

*Forest sites $^{239+240}$Pu activities and inventories*

The mean inventory (i.e. sum of L, O and mineral horizons $^{239+240}$Pu activities) of all forest sampling sites is 41.3 Bq m$^{-2}$, whereas the measured $^{239+240}$Pu activity in the Kahuzi Biega forest is somewhat smaller compared to the Kibale and Nyungwe forests (Fig. 2; Kahuzi Biega: 32.7 Bq m$^{-2}$, Kibale: 42.9 Bq m$^{-2}$; Nyungwe: 48.4 Bq m$^{-2}$). The majority of L horizon samples fall below the $^{239+240}$Pu detection limit (0.01 Bq kg$^{-1}$; 40 out of 55 samples). O horizon samples show distinctively higher $^{239+240}$Pu activities (Fig. 3). However, the contribution of the O horizons to the $^{239+240}$Pu inventory of the soil profile is small (mean: 1.2%), because of low bulk density (approx. 0.2 Mg m$^{-3}$) and O horizon thickness (mean 5 cm). Few (13%, 7 of 55 samples) of $^{239+240}$Pu activities of the 0 to 60 cm depth mineral soil layers fall below the detection limit, while the $^{239+240}$Pu activities of samples within the detectable range are at least three times higher than the detection limit (Fig. 2). In contrast, almost no activity is detected in the deeper soil layer from 60 to 120 cm.

Comparing the $^{239+240}$Pu activities at different topographic positions does not result in a consistent $^{239+240}$Pu activity to topography relation (Fig. 3). While the foot-slopes in Rwanda show the highest $^{239+240}$Pu activities, the opposite is the case for foot-slopes in Uganda and DR Congo. At the plateau sites in Uganda and Rwanda, a lower $^{239+240}$Pu activity compared to the slope sites is found.

A large $^{239+240}$Pu activity standard deviation of the forest sites was found for all topographic positions (Fig. 3). The mean $^{239+240}$Pu inventories found at the slope and foot-slope sites fall within the range of one standard deviation ± mean $^{239+240}$Pu inventory of the corresponding plateau sites. The only exception is the foot-slope in the Ugandan forest that falls in range by two standard deviations of the plateau mean.

*Cropland sites $^{239+240}$Pu activities and inventories*

At both cropland study sites, a distinctively lower $^{239+240}$Pu activity relative to the forest sites is found. The lowest activity of $^{239+240}$Pu is found at slope positions in DR Congo where 50% (n = 16) of sampling sites fall below the detection limit. The measurable slope samples show a mean and standard deviation of 0.019±<0.01 Bq kg$^{-1}$. A pronounced increase of the $^{239+240}$Pu activities can be observed at foot-slope positions with activity also detectable in the sampled 60-100 cm deeper soil layer (Fig. 3). Hence, the $^{239+240}$Pu activity at the DR Congo cropland site follows a topography related spatial pattern from low activities at slope to elevated activities at foot-slope positions.

In comparison to the cropland study site in DR Congo, the activities at the Ugandan cropland site are much higher (mean $^{239+240}$Pu activity at slope sites DR Congo: 0.012 Bq kg$^{-1}$, Uganda: 0.046 Bq kg$^{-1}$) and do rarely (3 of 44 samples) fall below the detection limit. Variability of $^{239+240}$Pu activities at Ugandan site is extremely high and shows for slope positions a coefficient of variation of 76%. In contrast to DR Congo cropland, lower $^{239+240}$Pu activities are found at Ugandan foot-slope sites compared to slope positions (Fig. 3). Furthermore, the Ugandan foot-slope positions showed almost no $^{239+240}$Pu activity in the deeper soil layer of 60-100 cm soil depth (Fig. 3).

DR Congo plateau sites (Ref$_{plt}$), assumed to represent the preserved full inventory of the global fallout, show a substantially lower $^{239+240}$Pu inventory compared to the about 30 km apart located forest sites (DR Congo Ref$_{plt}$: 8.0 Bq m$^{-2}$ vs. mean forest Ref$_{for}$: 32.7 Bq m$^{-2}$). Similarly, Uganda cropland plateau sites $^{239+240}$Pu inventory (24.4 Bq m$^{-2}$) are lower compared to nearby (~10 km) forest sites, but with a lower relative difference compared to their DR Congo counterparts (relative difference Ref$_{for}$ vs. Ref$_{plt}$: Uganda 43%, DR Congo 75%). The $^{239+240}$Pu activity below the 0 - 20 cm topsoil layer at the plateau sites in DR Congo show a sharp reduction of the $^{239+240}$Pu activity in deeper soil layers (Fig. 3), while at the Ugandan cropland sites, soil layers down to 40 cm show significant $^{239+240}$Pu activity.

*Cropland soil erosion and sedimentation*

An important piece of information that is provided by the erosion module of the mass balance model is the minimum quantity of soil loss that is required to cause a reduction of the $^{239+240}$Pu inventory that falls below the detection limit after the model integration period of 55 yrs. The difference between the Ref$_{for}$ and the Ref$_{plt}$ $^{239+240}$Pu baseline reference leads to substantial

differences in modelled erosion. The model indicates that at the DR Congo cropland sites, soil loss of at least 37 cm (55 yrs.)$^{-1}$ is necessary before the $^{239+240}$Pu activity falls below detection limit using Ref$_{for}$, while using Ref$_{plt}$ at least 10 cm (55 yrs)$^{-1}$ of soil loss must have taken place before the detection limit is reached. At the Ugandan cropland sites, a $^{239+240}$Pu inventory reduction to reduce activity below detection limit is found for 43 cm soil loss (55 yrs.)$^{-1}$ when applying Ref$_{for}$ and 32 cm (55 yrs)$^{-1}$ when applying Ref$_{plt}$, respectively (Fig. 4).

Also, a pronounced sensitivity of the mass balance model on the tillage depth parameter is found. A deviation from an assumed 20 cm plough depth of ±5 cm causes a change of the required soil loss until the detection limit is reached of about ±24%. Testing the low frequency but high magnitude soil erosion scenario (only 5 extreme erosion years within 55 years simulation period), showed that detection limit is reached already at 81% of soil loss compared to the continuous 55 years erosion rate. Hence, the sensitivity of the ±5 cm plough depth exceeds the impact of the erosion year frequency, even for this extreme scenario assumption in the NiCo ecosystem (approx. 20 erosive rainfall events per rain season).

The number of sampling sites that are considered to be subject to sedimentation is strongly controlled by the assumption on the applied reference (Ref$_{for}$ and Ref$_{plt}$). When using Ref$_{for}$, 4 slope sites (DR Congo: 0, Uganda: 4) show sedimentation greater than 5 cm (55 yrs)$^{-1}$. This number increased to 24 sites following Ref$_{plt}$ (DR Congo: 9, Uganda: 15). For both reference scenarios, sedimentation at slope positions is much lower in DR Congo as compared to Ugandan cropland (Fig. 5). The foot-slope sites in DR Congo show distinctively higher $^{239+240}$Pu inventories compared to the slope sites (Fig. 5). However, the mean inventory of foot-slope positions is still lower than the Ref$_{for}$ $^{239+240}$Pu inventory (28 vs. 32 Bq m$^{-2}$), which would be interpreted as an indicator for weak soil erosion considering Ref$_{for}$ for soil redistribution calculation. In contrast, if Ref$_{plt}$ is used in the calculation, the same foot-slope positions would be interpreted as sites that received substantial sedimentation exceeding 40 cm (55 yrs)$^{-1}$ (Fig. 5). The foot-slope sites in DR Congo show a pronounced $^{239+240}$Pu activity in many samples of deeper soil layers (60-100 cm), while no such deeper soil layer $^{239+240}$Pu activity is found at foot-slope sites of the Ugandan study site (Fig. 3). Simulated sedimentation that exceeds the site specific sampling depth minus the plough depth (i.e. 40 cm at the slope locations assuming a 20 cm plough depth) is rarely found for both reference assumptions (DR Congo Ref$_{plt}$: 1, Ref$_{for}$: 0; Uganda Ref$_{plt}$: 6, Ref$_{for}$: 2) and therefore suggests a limited impact of enrichment processes by selective transport of fine soil particles.

## 4. Discussion

*Applicability of $^{239+240}$Pu as soil erosion tracer in tropical Africa*

In this study, a $^{239+240}$Pu based soil redistribution analysis at three forest and two cropland sites in the NiCo region was carried out. It is shown that for the majority of samples the topsoil $^{239+240}$Pu activity is high enough to be successfully measured and provide insight on soil redistribution in tropical Africa over the past decades. To our knowledge, this is the first $^{239+240}$Pu based soil redistribution study in tropical Africa.

The $^{239+240}$Pu inventories found in this study are much higher than expected based on the global fallout estimates reported by Kelley et al. (1999) and Hardy et al. (1973) (4.8 Bq m$^{-2}$ and 11.1 Bq m$^{-2}$ for 10°N and 10°S). It is well established that the initial deposition of $^{239+240}$Pu fallout was strongly dependent upon latitude; the study of Kelley et al. (1999) compared $^{237}$Np, $^{239}$Pu and $^{240}$Pu inventories from soil cores collected from undisturbed, worldwide locations. One data point (Muguga, Kenya) included in the results of Kelley et al. (1999) yields a $^{239+240}$Pu inventory of 19.2 Bq m$^{-2}$, which is significantly lower than the inventories for soils in the mid-latitude regions of the Northern Hemisphere, e.g., Munich, Germany exhibits a $^{239+240}$Pu inventory of 104 Bq m$^{-2}$. Although the results of Kelley et al. (1999) are drawn from a limited set of samples, they nevertheless make the case for a strong dependency of inventory vs. latitude, as well as annual precipitation, as factors that control the depositional inventory. For the latitudinal classification of Hardy et al. (1973) only two measurements between 0° and 10°S were located in Africa (1.2°S Muguga, Kenya & 9.0°S Luanda, Angola; Hardy et al., 1973; Kelley et al., 1999). Both stations receive a substantially lower annual precipitation (960 and 430 mm for Kenya and Angola, respectively) than the NiCo region (>1400 mm yr$^{-1}$; Fick and Hijmans, 2017) and show contrasting $^{239+240}$Pu inventories of 19.2 Bq m$^{-2}$ in Kenya and 3.4 Bq m$^{-2}$ in Angola. Hence, it is not surprising to find higher baseline $^{239+240}$Pu inventories within the NiCo region than in Kenya or Angola. The three pristine forests show mean $^{239+240}$Pu inventories between 33 and 48 Bq m$^{-2}$ (DR Congo: 32.7±7.7 Bq m$^{-2}$; Uganda: 42.91±15.5 Bq m$^{-2}$; Rwanda: 48.4±18.2 Bq m$^{-2}$), which is sufficiently high for conducting soil redistribution studies. However, half of the slope sites (14 of 28) at the cropland site in DR Congo fall below the detection limit (0.01 Bq kg$^{-1}$; ~5 Bq m$^{-2}$). This is partly caused by the sampling design of this study, which is based on large and deep single soil increments that cover the soil depth from 0 to 60 cm and 60 to 100 cm at the slope and foot-slope positions. A straightforward way to increase the $^{239+240}$Pu activity in the sample is the reduction of the sample increment depth for a corresponding increase of the topsoil proportion that has a higher $^{239+240}$Pu activity (see 20 cm increments taken at cropland plateau sites Fig. 3). However, a reduction of the sampling increments necessarily requires an additional analysis of deeper soil layers in highly degraded soil systems, particularly in regions with complex soil redistribution patterns to include the full $^{239+240}$Pu inventory.

For future $^{239+240}$Pu based soil redistribution investigations in tropical cropland areas, a two-layered sampling scheme is suggested that individually samples the plough layer and the deeper soil layer (e.g. 0-20 cm and 20-60 cm at a slope and an additional 60-100 cm sample at foot slope sites). The two-layered sampling scheme accounts for the large number of samples falling below the detection limit and allows for a better understanding of the $^{239+240}$Pu depth distribution and thereby depletion status of the sediment source area. In tropical forests, the sampling scheme can be focused on mineral soil as the activity in the litter and organic layer does not substantially contribute to the $^{239+240}$Pu inventory. Sedimentation at the three forest study sites did not exceed 60 cm at the foot slope, suggesting that a sampling depth of 0-60 cm is sufficient to include the full inventory. Nevertheless, additional deeper soil sampling, with reduced increment depth (e.g. 20 cm) at foot slope locations, is suggested to account for different environmental conditions and potentially higher foot slope sedimentation. The determination of appropriate reference sites is critical. At plateau sites, three depth increments (0-20 cm; 20-40 cm; 40-60 cm) were sampled to understand the depth distribution of sites without substantial water erosion. At the plateau sites, no $^{239+240}$Pu activity was found in the 40-60 cm soil layer. Hence, a single sample that integrates soil from 0 to 40 cm depth seems to be sufficient. However,

the number of reference samples should be high enough to represent a robust mean with regards to $^{239+240}$Pu variability typically associated with the corresponding land use.

*$^{239+240}$Pu reference inventory*

In this study, two different reference scenarios are taken into account to address potential differences of $^{239+240}$Pu inventories between stable forest and cropland plateau positions: (i) Ref$_{for}$ mean of specific forest sites (ii) and Ref$_{plt}$ mean of cropland plateau positions of the specific sites. A high variability of $^{239+240}$Pu activities and corresponding inventories in three pristine tropical forests within similar topographic positions that exceeds differences in Pu inventories along slope positions is found (Fig. 3). The variation of forest $^{239+240}$Pu inventories due to bioturbation and fallout infiltration patterns (e.g. caused by throughfall or stem flow patterns) exceeds a potential soil redistribution impact, which is illustrated by the standard deviation of the plateau sites that covers the variability of the slope and foot-slope composite samples (Fig. 3). Additional evidence that soil redistribution processes in the studied forest systems are low, is that no major differences between chemical and physical soil properties are found along geomorphic gradients (Reichenbach et al., 2021). This finding is in line with global erosion plot studies from tropical forest plots (mean erosion 0.2 Mg ha$^{-1}$ yr$^{-1}$, 39 plots with 116 plot years; Xiong et al., 2019). Observations on sediment delivery monitoring in the NiCo region show that the amount of sediment delivery from pristine forests is typically less than 1 Mg ha$^{-1}$ yr$^{-1}$ (personal communication with Simon Baumgartner, UCL runoff monitoring FORSEDCO project). Furthermore, Drake et al. (2019) exemplarily showed in the NiCo region that particular matter export within pristine tropical forest catchments are dominated by organic matter export with little to no mineral sediment being transported. In contrast, partly deforested catchments with agricultural use show substantial carbon delivery by organo-mineral complexes that indicates detachment and transport of the mineral soil layers, which is again in line with the soil erosion results of this study (Drake et al., 2019). Hence, the forest sites are assumed to represent almost the entire $^{239+240}$Pu inventory of the baseline inventory without substantial loss due to soil erosion. The basic assumption behind the reference sites is that the full inventory is preserved as no soil redistribution has taken place and the $^{239+240}$Pu inventories of both Ref$_{for}$ and Ref$_{plt}$ are supposed to be similar. However, the mean cropland plateau $^{239+240}$Pu inventory in Uganda is about half (24.4±7.6 Bq m$^{-2}$) and in DR Congo only a quarter (8.0±1.0 Bq m$^{-2}$) of the mean inventories found in the nearby (<30 km) pristine forest sites (Fig. 2), which cannot be explained by local rainfall heterogeneities and corresponding spatial patterns of the baseline inventory. Deeper soil layers below 60 cm depth at cropland foot-slope positions show $^{239+240}$Pu activity, which is a clear proxy for substantial sedimentation (Fig. 3). However, the $^{239+240}$Pu inventory of these locations are not exceeding the forest reference inventory Ref$_{for}$ and would therefore be interpreted as weak soil erosion applying the mass balance model. This is unexpected and can point at a variety of different processes at play that were not investigated by this study. The measured $^{239+240}$Pu activity at the foot-slope positions may underestimate the $^{239+240}$Pu inventory due to three potential processes: (i) Sedimentation exceeds the sampling depths that does not include the full inventory, (ii) deposition of sediments that are already depleted in $^{239+240}$Pu as the inventory at the eroded source area has been subject to pronounced soil and corresponding $^{239+240}$Pu loss and (iii) sedimentation sites may temporarily turn into erosion sites. However, an indication for these processes would be an increasing

$^{239+240}$Pu activity with soil depth, which is not reflected in the data (Fig. 3). Another potential explanation is that $^{239+240}$Pu inventories are reduced due to plant uptake and subsequent plant harvest. However, a substantial plant uptake by crops, like that observed for the fallout radionuclide $^{137}$Cs (White and Broadley, 2000; Zhu and Smolders, 2000), is unlikely as no elevated

$^{239+240}$Pu activity was found in harvested crops of other studies (Akleyev et al., 2000). Another potential pathway of soil and $^{239+240}$Pu leaving the cropland plateau sites is harvest erosion associated to commonly cultivated root crops (i.e. cassava, sweet potato, groundnuts). In temperate regions, harvest erosion rates up to 12 Mg ha$^{-1}$ yr$^{-1}$ have been reported for different crop types (potato: 2.5 - 6 Mg ha$^{-1}$ yr$^{-1}$; Auerswald and Schmidt, 1986; Belotserkovsky and Larinovo, 1988; Ruysschaert et al., 2007) (sugarbeet: 5 - 8 Mg ha$^{-1}$ yr$^{-1}$; Auerswald et al., 2006) (chicory: 8.1 - 11.8 Mg ha$^{-1}$ yr$^{-1}$; Poesen et al., 2001). With cassava

and sweet potato being the main food crops within the NiCo region (cassava has a higher proportion on the less fertile soils in the DR Congo, while more sweet potato is cultivated in Uganda), this is a likely source of reduction of $^{239+240}$Pu inventories. To illustrate the potential effect of harvest erosion, a simple example shows that 5.5 Mg ha$^{-1}$ yr$^{-1}$ of sediment delivery would roughly cause a 20% reduction of the baseline reference over 55 years (assuming a 20 cm plough depth and 1.35 Mg m$^{-2}$ bulk density). Harvest erosion can be assumed as a process that has a limited spatial distribution as long as the land use and crop

yields are not causing pronounced spatial patterns. Therefore, in systems where harvest erosion is a relevant driver of $^{239+240}$Pu export, Ref$_{plt}$ would be the valid reference for soil redistribution estimations. However, an accurate estimation of the contribution of $^{239+240}$Pu loss due to harvest erosion since the 1960s is impossible as limited information is available on soil harvesting loss of cassava and potato by hand cultivation. Therefore, both Ref$_{for}$ and Ref$_{plt}$ are taken into account within this study to cover the range from a fully preserved to a depleted $^{239+240}$Pu reference inventory in the study sites.

*Soil redistribution in cropland of the NiCo region*

Both cropland sites show indications of soil redistribution. Particularly the cropland study site in DR Congo shows evidence for (i) soil loss due to a high number of slope samples with activities falling below the detection limit (50%) and (ii) sedimentation as evidenced by a clear $^{239+240}$Pu fingerprint in the deeper soil layers of the foot-slope samples. Compared to the

375 Ugandan cropland, the DR Congo cropland shows a much stronger difference between $^{239+240}$Pu inventories of slope positions and Ref$_{for}$ (Fig. 2). We relate this discrepancy of the different duration since DR Congo and Uganda tropical forest has been converted into cropland. Forest into cropland conversion at the Ugandan study site took place during the 1970's. Hence, the area was under arable use for about 40 years compared to 55 years (since the test ban treaty) at the DR Congo study site. Therefore, the Ugandan cropland was exposed to soil erosion for a roughly 27% shorter time compared to DR Congo cropland.

However, the relative $^{239+240}$Pu inventory reduction at slope sites in Uganda is about 29% compared to Ref$_{for}$, while in DR Congo the relative reduction is 83%. The much stronger relative $^{239+240}$Pu reduction in DR Congo cannot be just explained by the shorter cropland cultivation period of the Ugandan cropland site. In direct comparison between the two sites, no major difference regarding slope steepness (12°-13° in both study sites) and rainfall erosivity (Fenta et al., 2017) was observed. The different land use during the main time of atmospheric nuclear weapon tests (DR Congo=cropland; Uganda=forest) might have

caused higher spatial variability of the initial $^{239+240}$Pu inventory associated with complex infiltration (e.g. stem flow) and

troughfall patterns in forests (Hofhansl et al., 2012). However, these small-scale variation is likely to be homogenised after 40 years of lateral soil movement by plough-based arable use. The main cultivated crops differ between the Ugandan and DR Congo study sites substantially. In Uganda sweet potato and maize are the major crops, while DR Congo cropland is dominated by cassava. This difference may have an impact on water and harvest driven soil erosion processes.

The determined mean soil redistribution rates at cropland slopes found in this study (negative values indicate net erosion and positive values net sedimentation; DR Congo: -51.4 Mg ha$^{-1}$ yr$^{-1}$ Ref$_{for}$, -1.4 Mg ha$^{-1}$ yr$^{-1}$ Ref$_{plt}$; Uganda (referred to 40 yrs. of arable land use and corresponding soil redistribution): -18.4 Mg ha$^{-1}$ yr$^{-1}$ Ref$_{for}$, 27.8 Mg ha$^{-1}$ yr$^{-1}$ Ref$_{plt}$) show contrasting results with respect to the assumed reference (Fig. 5). In comparison to values observed (Boardman and Poesen, 2006) and modelled globally (Borrelli et al., 2017), the high erosion Ref$_{for}$ simulations are in good agreement with plot monitoring results

observed in the region (mean soil loss of 68.2 Mg ha$^{-1}$ yr$^{-1}$; Lewis and Nyamulinda, 1996). Over very short distances, high soil redistribution heterogeneities ranging from sedimentation to heavy soil loss are found, which might be an effect of smallholder farming structures. These farming structures are potentially mitigating soil loss rates in the NiCo region due to decreasing hydrologic connectivity (Nunes et al., 2018; Baartman et al., 2020) along slopes due to a high degree of "patchiness" and a large number of field boundaries (mean field size 450 m²; Fig. 5). Any conversion of this smallholder farming structure into

large scale farming structures, as known from regions with mechanised agriculture, will have devastating effects on soil degradation rates in the region.

## 5. Conclusions

This study demonstrates the feasibility of analysing fallout radionuclides $^{239}$Pu and $^{240}$Pu as a tool to assess soil degradation processes in tropical Africa. Interpreting $^{239+240}$Pu activity and inventories in soils and organic layers, we assessed soil

redistribution rates along three pristine forests catenae and in two cropland catchments in the White Nile-Congo rift region. $^{239+240}$Pu inventories in forest did not follow a topography related distribution, which points at minor soil erosion. In contrast, cropland sites show signs for substantial soil erosion and sedimentation that exceeds 40 cm over a period of 55 years. However, the selection of an appropriate reference is critical due to a potential $^{239+240}$Pu inventory reduction associated with cropland use other than water erosion. Very high intra-slope variability of the $^{239+240}$Pu inventories in cropland was found (coefficient of

variation up to 67%) with sites of pronounced sedimentation in close distance to highly eroded sites, potentially a result of soil cover dynamics due to smallholder farming structures with small fields and individual management. Keeping smallholder farming structures active is essential to mitigate soil degradation in the region, also under current agricultural intensification efforts. Particularly in regions with limited infrastructure and challenging monitoring conditions, $^{239+240}$Pu based soil redistribution analysis can shed light on the pace of soil degradation, which remains a major challenge for future food security

in tropical Africa.

**Data availability**

The data is available within the project TropSOC database publication:

Doetterl, S., Asifiwe, R. K., Baert, G., Bamba, F., Bauters, M., Bukombe, B., Cadisch, G., Cizungu, L., Cooper, M., Hoyt, A., Kabaske, C., Kalbitz, K., Kidinda, K. L., Maier, A., Mainka, M., Mayrock, J., Muhindo, D., Mujinya, B., Mukotanyi, S. M., Nabahungu, L., Reichenbach, M., Rewald, B., Six, J., Stegmann, A., Summerauer, L., Unseld, R., van Oost, K., Verheyen, K., Vogel, C., Wilken, F., and Fiener, P.: Organic matter cycling along geochemical, geomorphic and disturbance gradients in forests and cropland of the African Tropics - Project TropSOC DATABASE_v1.0., Earth System Science Data Discussions, https://doi.org/10.5194/essd-2021-73, 2021a.

Doetterl, S., Bukombe, B., Cooper, M., Kidinda, L., Muhindo, D., Reichenbach, M., Stegmann, A., Summerauer, L., Wilken, F., and Fiener, P.: P. TropSOC Database. Version 1.0, GFZ Data Services, https://doi.org/10.5880/fidgeo.2021.009, 2021b.

**Author contribution**

This paper represents a result of collegial teamwork. FW, SD, PF and KvO developed the study design. $^{239+240}$Pu activity analysis was carried out by MK. DM carried out the field campaign. Data processing and illustration was carried out by FW. FW, SD, PF, KVO and KM contributed to data analysis and interpretation. FW drafted the manuscript, while all authors reviewed and approved the final version.

**Competing interest**

The authors declare that they have no conflict of interest.

**Acknowledgements**

We gratefully acknowledge the outstanding commitment for project TropSOC and fieldwork support of Landry Cizugu, Benjamin Bukombe, Laurent Kidinda, Mario Reichenbach and for labwork Anna Stegmann, Julia Mayrock, Moritz Mainka and Robin Unseld.

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

 **Tables**

Table 1: Coordinates and main characteristics of investigated study sites. Mean annual rainfall and temperature data is based on Fick and Hijmans (2017).

| Study site | Location [Lat., Long., Elevation] | Max. slope [°] | Mean annual rainfall [mm] | Mean annual temperature [°C] | Mean field size [m²] |
|---|---|---|---|---|---|
| DRC forest | 2°19'S, 28°44'E, 2208-2248 m | 60 | 1925 | 15.2 | - |
| Uganda forest | 0°28'N, 30°22'E, 1271-1306 m | 55 | 1277 | 20.7 | - |
| Rwanda forest | 2°28'S, 29°06'E, 1271-1306 m | 60 | 1689 | 17.3 | - |
| DRC cropland | 2°35'S, 28°43'E, 1711-1747 m | 17 | 1586 | 17.7 | 662 |
| Uganda cropland | 0°34'N, 30°20'E, 1470-1510 m | 20 | 1459 | 19.5 | 336 |

575

Table 2: Numbers of samples taken at three forest (DR Congo, Uganda, Rwanda) and two cropland (DR Congo, Uganda) study sites. While the O horizon depth is measured individually for each sample, the L horizon depth is assumed to be 1 cm as it was not possible to be accurately measured. L = L horizon; O = O horizon; M = mineral layer 1: 0-60 cm and 2: 60-120 in forest and 60-100 cm in cropland.

| | Forest | | |
|---|---|---|---|
| | Plateau | Slope | Foot-slope |
| L, O, M1 | 10 | 6 | 6 |
| M2 | 0 | 0 | 6 |
| | Cropland | | |
| | Plateau* | Slope | Foot-slope |
| M1 | 12 | 27 | 12 |
| M2 | 0 | 0 | 12 |

*Three depth increments 0 - 20 cm, 20 - 40 cm, 40 - 60 cm

580

**Figures**

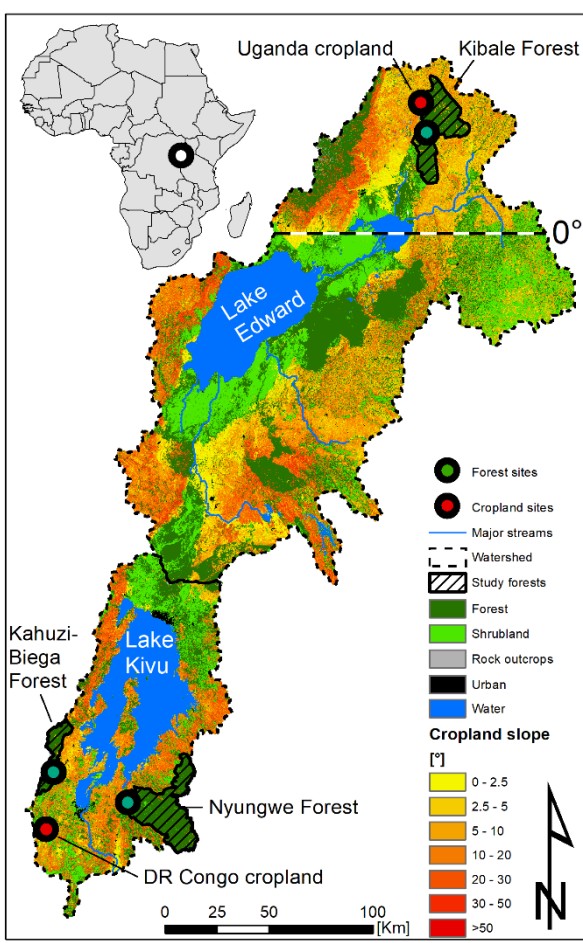

Figure 1: Study area, locations of forest and cropland sites, topography and land use in the White Nile-Congo rift region (land use data is based on: ESA Climate Change Initiative - Land Cover project 2017).

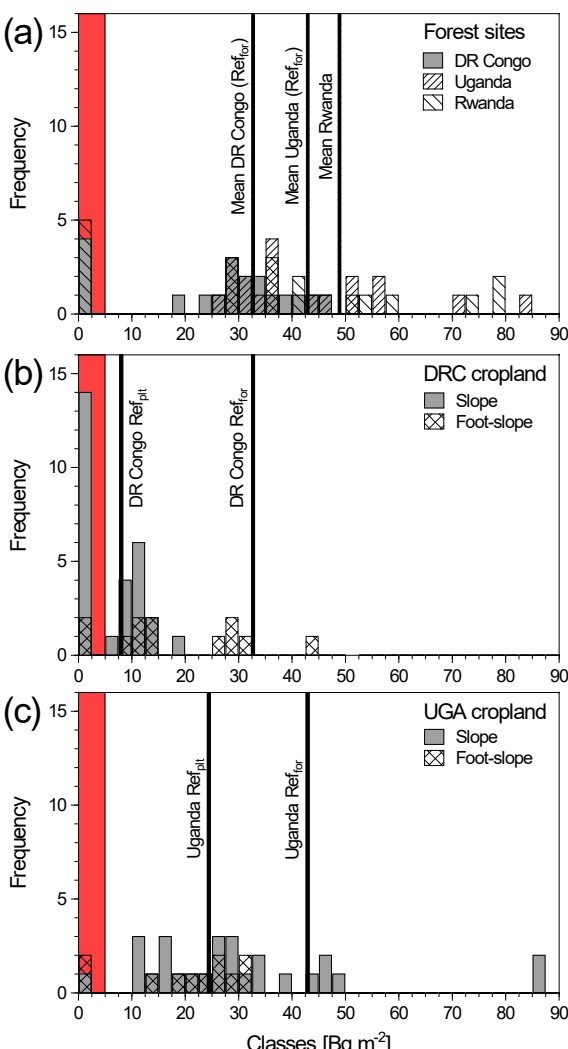

Figure 2: Distribution of $^{239+240}$Pu inventories in three forests and two cropland study sites. In (a) the $^{239+240}$Pu histogram for three forest sites is given, while (b) and (c) represent the cropland distribution at the slope and foot-slope topographic positions. The vertical lines represent the mean values serving as forest reference (Ref$_{for}$) and cropland plateau reference (Ref$_{plt}$). Red section at the left of the graphs illustrates the detection limit.

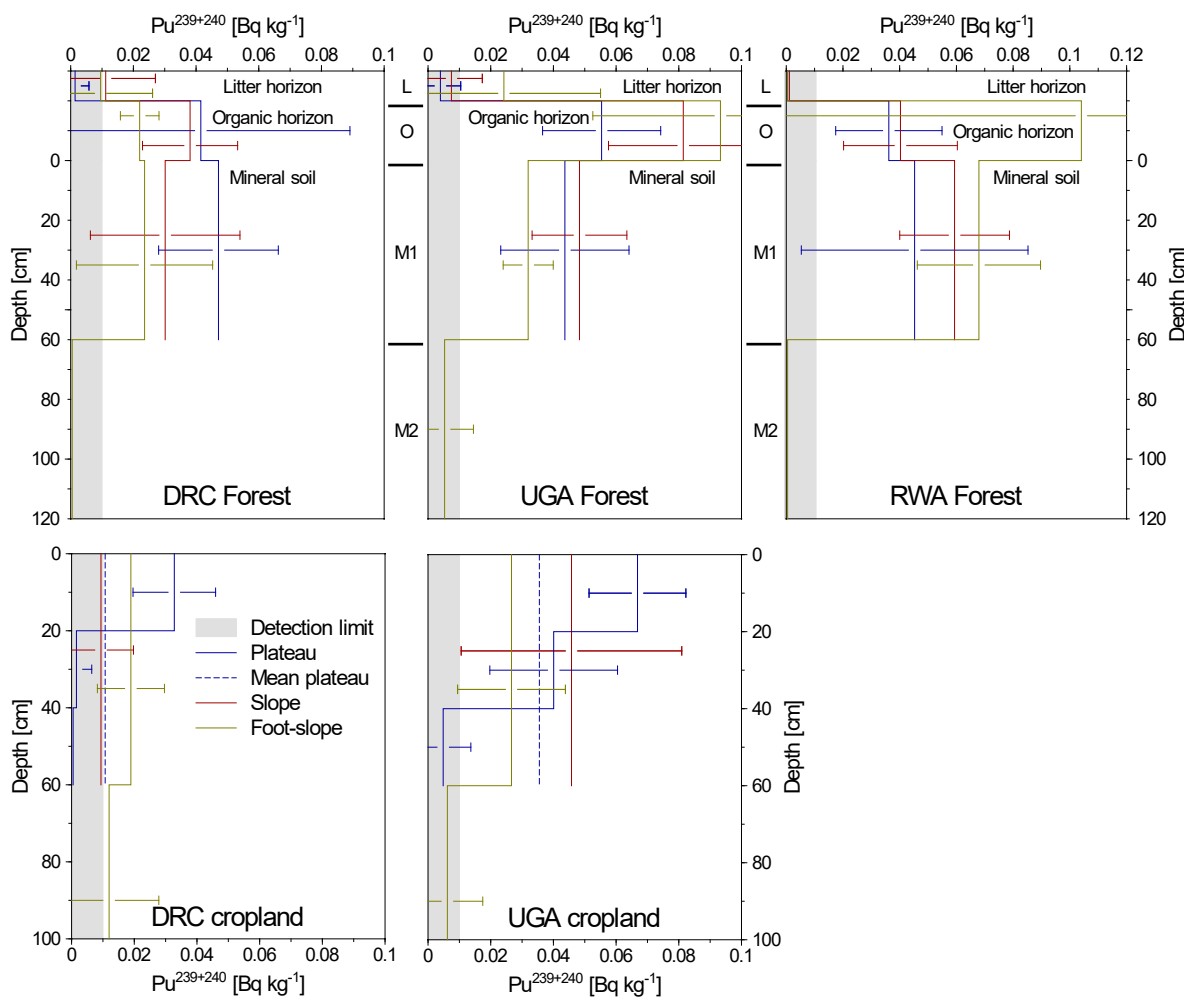

Figure 3: Catenae related depth profiles of $^{239+240}$Pu activity and standard deviations (whisker) within three pristine forests (DRC: DR Congo, UGA: Uganda, RWA: Rwanda) and two cropland study sites. Please note that the L and O horizons are for illustrational purposes shown thicker than they naturally appear.

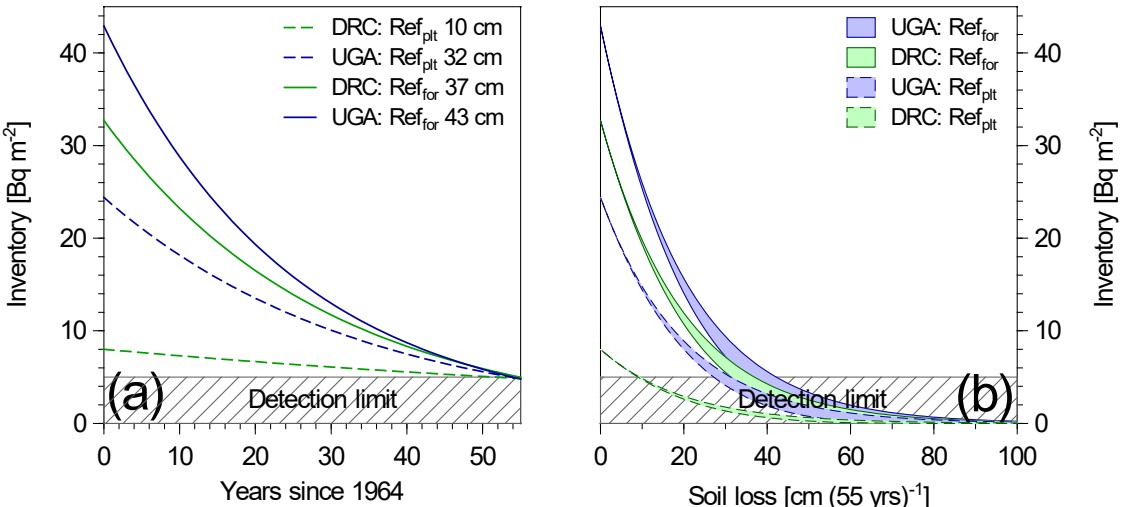

Figure 4: DR Congo (DRC) and Uganda (UGA) cropland soil erosion results of the mass balance model. A scenario assessment was carried out to understand $^{239+240}$Pu inventory reduction until detection limit is reached after the simulation period: (a) soil erosion magnitude and corresponding (b) soil erosion frequency according to the mean forest (Ref$_{for}$) and mean cropland plateau (Ref$_{plt}$) assumption. (a) Represents the minimum quantity of soil erosion to pass the detection limit with different reference $^{239+240}$Pu inventories. All displayed curves show the corresponding scenario runs of minimum soil erosion to cause a $^{239+240}$Pu inventory reduction that falls below the detection limit. (b) Comparison between a continuous 55 years soil erosion rate vs. 5 erosive years that cause the same quantity of soil erosion. Soil erosion from 1 to 100 cm (55 yrs)$^{-1}$ is simulated. The coloured area describes the difference between the two scenarios, where the lower line is the 5 year scenario.

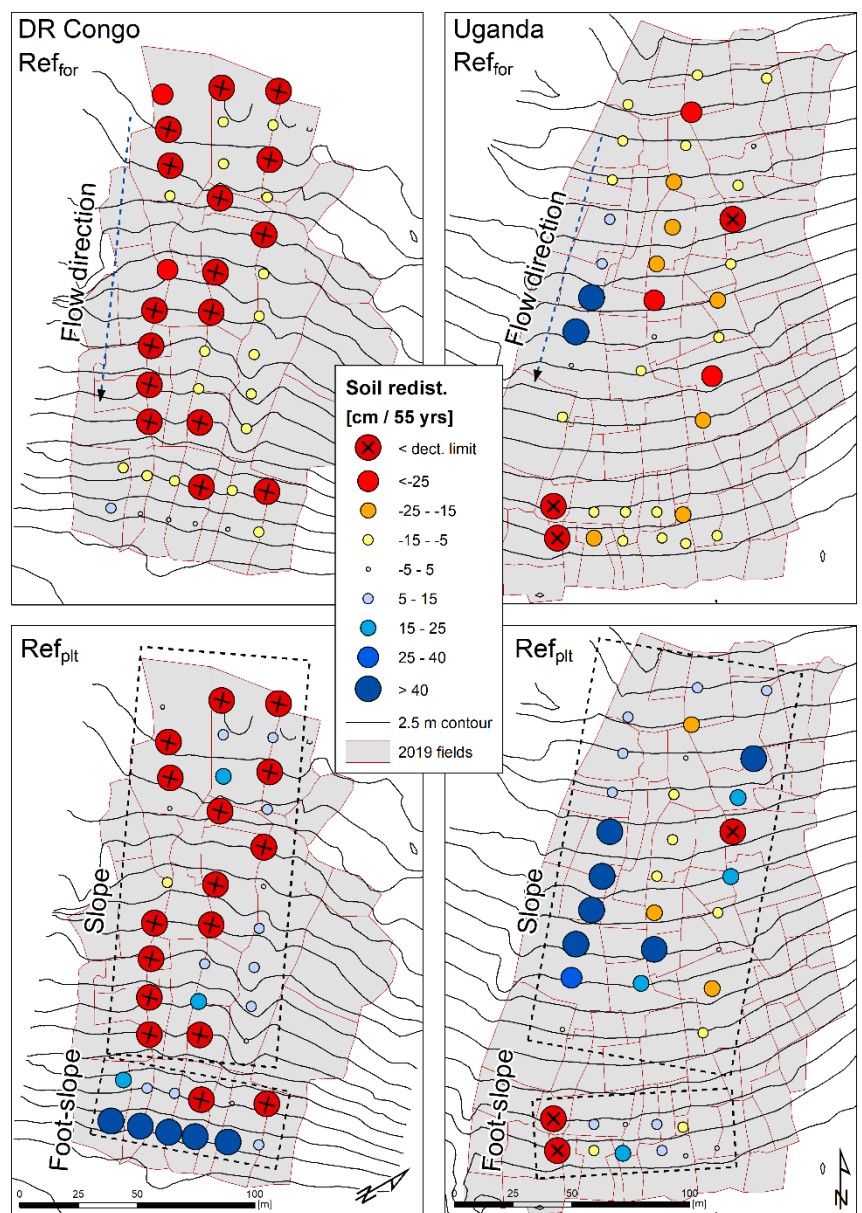

Figure 5: Soil redistribution derived from $^{239+240}$Pu measurements using a mass balance model. Two different reference inventories (upper: $Ref_{for}$ = mean forest reference; lower: $Ref_{plt}$ mean cropland plateau reference) in two cropland areas of the White Nile-Congo rift region (left: DR Congo, right: Uganda) were used for calculation.

**Appendix A**

Figure A1: Overview photos and orthomosaics of the DR Congo (left) and Uganda (right) cropland sites. Image acquisition dates: DR Congo (left panel) overview photo Sep. 1st, 2018, orthomosaic Sep. 14th, 2019; Uganda (right panel) overview photo Oct. 12th, 2019, orthomosaic Jun. 12th, 2019.

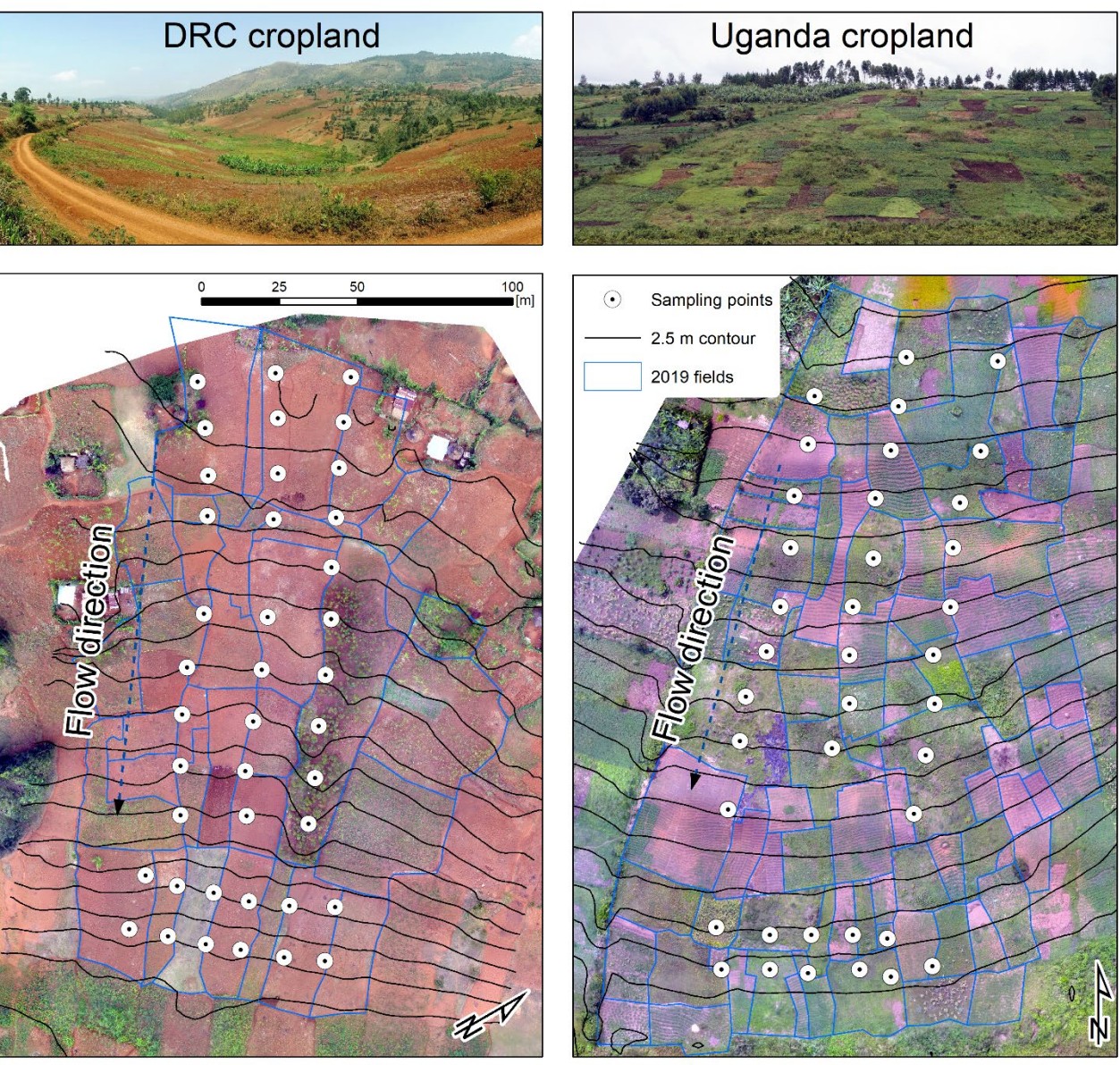