# Peer review of "Assessing soil redistribution of forest and cropland sites in wet tropical Africa using 239+240Pu fallout radionuclides"

_SOIL, 2020_

## Referee Comment (RC1) · Anonymous Referee #1 · 15 Jan 2021

This manuscript presents a very interesting study proposing to use plutonium inventories (as a surrogate to cesium) to calculate soil redistribution rates during the post-1960 period at five different sites (including cropland and forest sites) in tropical Africa. This is one of the first studies (if not the very first study) using fallout radionuclides to reconstruct soil erosion/accumulation processes in this region of the world, and it is therefore of large interest to the scientific community in general, and the audience of SOIL journal in particular.

Overall, the study was well designed, and the results are well described using nice figures. However, the text should be clarified at some places and several improvements

could further increase the potential impact of the manuscript (see the detailed comments below). Therefore, I think that major revisions are required before the potential final acceptance of the manuscript for publication.

Major comments

In the Methods, adding a table comparing the main characteristics of the 5 investigated sites would be very helpful for the readers. For instance, text on page 4 is hard to follow and an additional table would definitely help comparing the site characteristics at a glance.

On L.116, you refer for the 1st time to "subsoil" (which then appears on many occasions in the text). In my opinion, this is misleading as you are referring to footslope locations (you even refer to "colluvial sites" in the text) so that – in my opinion – you expect to find deep accumulations of eroded soil at those locations which likely mainly consist of "eroded topsoil" which may be progressively buried at the footslope » you should rather refer to "deeper soil layers" and avoid using the "subsoil" term. . . This should be corrected throughout the entire manuscript.

Regarding the calculation of soil redistribution rates, I wondered when reading section 2.4 how you took the reference inventories into account, then when reading section 2.5, I likely found the answer: you use it as an important factor in the sensitivity analysis given the large potential uncertainties, right? Do you confirm this?

In the discussion, on pages 10-11, in my opinion, there is one hypothesis lacking to explain the low inventories measured in this study. What about the potential export of soil/plutonium from the hillslope to locations located further downstream? Are the hillslopes connected to lower locations, how is the connectivity between the hillslopes and lower zones? We may imagine that part of the eroded soil and the associated plutonium have been exported from the hillslopes during the last several decades.

Furthermore, on LL.272-280, maybe in addition to rainfall depth, it is important to take

into account the latitude of the investigated sites, as fallout radionuclide inventories were showed to be strongly latitude-dependent (the location in the Northern or the Southern Hemisphere is also of importance). Finally, in the discussion, maybe adding a sequence on recommendations for future studies using fallout radionuclides in tropical environments would be useful. For instance, the results described on LL.205-209 confirm that, in future, the analysis of L/O horizon samples could be avoided because they contain only negligible proportions of the total plutonium inventories (this would save time to analyse more soil depth increments for instance).

Detailed remarks Title

Maybe the title could be slightly improved, for instance: "assessing soil redistribution at forest and cropland sites. . ." (as you quantified soil erosion and accumulation?)

Abstract

L.18 (and all throughout the manuscript, I didn't list all the word occurrences): conversion INto arable land (instead of in)

L.18 unclear what you mean with "challenging local conditions"

L.21 "a relatively high inventory" » as this is all relative, maybe you could adapt the phrasing; e.g. this is definitely not high compared to the inventories observed in other regions of the world

L.23 "up to 37 and 40 cm" » maybe provide the soil redistribution rates instead ?

L.25 "insight into" instead of "insight on"

L.26 Does world deserve a capital letter here?

Introduction L.33 "low soil cover conditions" » sparse vegetation cover of the soil?

L.35 maybe specify that you refer to 'CROP yield'

L.35 'goes hand in hand' » maybe consider rephrasing?

L.37 for THE entire Sub-Saharan Africa?

LL.40-41 "to assess new non-degraded soils" » unclear, please rephrase

L.42-43: "the onset of soil erosion", maybe specify "at previously unaffected/pristine sites"?

L.45 "loss of potential reforestation" » impossibility of reforestation?

L.48 at suitable locations?

L.50 is detailed information > requires detailed information?

L.50 please avoid the repetitive use of 'specific'

L.54 "but are important" » although they remain crucial?

L.55 insight of > into

L.55 "internal soil redistribution dynamics" » do you mean at the hillslope scale?

L.58 "overcome by fallout radionuclides" » by the analysis of fallout radionuclides?

L.60 on > into

L.64 of the 137Cs activity until today?

L.64 in tropical and equatorial regions?

L.65 to much lower fallout

LL.66-67 extreme erosion rates. . . » there seems to be a verb missing in this sentence

L.67 Over the past decade. . . » there is a transition missing from the previous sentence here, in my opinion. Maybe add "To overcome these analytical difficulties. . ."?

L.69 their long half-life?

L.69 "without relevant decay" » this is not needed and should be removed

L.73 provide important insights into. . .

L.74 "where, to our best knowledge,. . ." » and, to the best of our knowledge, none was conducted. . .

LL.76-78 this should be rephrased in my opinion (strange to start with "we follow two major aims"; "exemplarily analysing" does not read well. . .)

Methods L.81 Lake Eduard » Edward?

L.83 conversion of forest into cropland?

L.85 gully > gullies

L.86 sum rainfall? I would remove 'sum"

L.87 maybe add the range of years during which this precipitation was measured?

L.87 subdivided into two cycles of wet and dry seasons?

L.88 storm events with large rainfall amounts?

L.92 throughout > across?

L.96-97 1950s / 1970s

L.99 "soil cover conditions are very patchy" » please rephrase

L.100 "in direct proximity" » not sure how Fig. 1 really illustrates this issue as I guess that the pixel size on Fig. 1 is not compatible with intra-field heterogeneities within the study sites?

L.110 "in order to understand variation of radionuclide inventories at sites" » do you mean "in order to quantify spatial variations in reference fallout inventories"?

L.112 L and O horizons

L.122 "from the study slope" » unclear what you mean here

L.171 "different nature" » unclear what you mean here

L.175 processes

L.191 from the literature, i.e. 80 kg m-3

Results (please avoid the use of the term "subsoil" in this section)

L.204 it seems that Fig. 3 is cited in the text before Fig. 2, maybe their sequence should be reversed.

L.208 rarely » maybe indicate the % instead?

L.210 Why do you consider subsoil as the layer lower as 60 cm?

L.214 within each forest site?

L.216 'that falls in range by two standard deviations of the plateau mean' » unclear what you mean here, please rephrase

LL.219-224: I have the impression that part of the text here is repeated from the previous paragraphs, please check and avoid repetitions

L.221 "0.019±0.006 Bq kg-1." » can we really be confident with 3 decimal digit significance here?

L.224 at slope > upslope?

L.233 "similar" » similarly?

L.244 and in contrast . . . using. . . » please rephrase

L.249 "testing the concentrated scenario" » this is probably not the best wording (extreme rainfall scenario?)

L.250 "after 19% less total soil loss"» confusing, please rephrase

L.253 "widely" > strongly?

L.256 "sloping positions" » unclear

L.256 weaker > lower?

L.261 'subsoil" » deep soil layers?

L.265 selective transport: are you referring to particle size here?

Discussion L.268 "within this study" > in this study?

L.272 "inventory findings" » inventory is found?

L.280 high for conducting soil redistribution studies

L.287 to cover » to include?

L.290 "within this study" > in this study?

L.296 small > low?

L.305 "represent almost the entire 239+240Pu inventory of the global fallout" » are you referring to the reference/baseline inventories here?

L.310 corresponding fallout patterns » are you referring to their heterogeneities in particular here?

L.313 at play not investigated by this study » that were not investigated by this study

L.315 "in subsoil" > with depth

L.317 like that observed for the fallout. . .

L.318 activity in crops » it is particularly very unlikely to find high Pu activities in vegetation 60 years after the fallout. . .

L.330 on the contribution » of the contribution?

L.337 falling below. . . » with activities falling below. . .

L.340 "varying length" » do you mean "duration" here? Or the period since the conversion of tropical forest into cropland?

L.342 "55 years" » shouldn't this 55-yr period be adapted depending on the duration since the conversion of forest into cropland?

L.345 "cropland use" » cultivation period?

L.347 were » was?

L.349; what about the occurrence of crop rotations in the different zones of interest?

L.354 "within the region" » observed in the region?

L.355 "The range of observed values at slopes spans from net sedimentation to heavy soil loss in direct proximity to each other" » unclear, please rephrase

Conclusions L.362 usability » feasibility of using/analysing?

L.364 catena > catenae?

L.365 "indicative for little to no soil erosion" » which demonstrates the (almost) absence of erosion?

L.367 "However, the selection of an appropriate reference is critical due to a potential 239+240Pu inventory reduction by harvest erosion in root crop dominated cropland systems." » it seems to go pretty far in the interpretation here, focusing on the magnitude of root crop erosion, which has not been quantified in the current research; I wonder what would be the importance of sediment export from the hillslope (see comment above)

Figures (overall, your figures are beautiful, congratulations for that!)

Fig. 1 maybe add the latitudes on this map (at least the Equator should be added); what is the source of the land use data?

Fig. 3, caption L.531 » illustrates

Fig. 4, caption L. 538: "limit of . . . " » limit with different. . .?

Fig.5, caption L. 546: were analysed » were used for calculation?

Tables

Table 1; it may seem counter-intuitive to analyse depth increments on the plateau and not at the footslope where sediment accumulates?

Caption L. 513: why not in Rwanda?

---

## Referee Comment (RC2) · Anonymous Referee #2 · 16 Jan 2021

First, I want to congratulate the authors with the very interesting and thorough study. The paper discusses the urgent problem of soil erosion which seriously impacts the region and there is still a huge lacuna of understanding on its dynamics. The topic is very suitable for the journal and the technique is novel in its geographical context. In that specific context, there is one major issue with the technique which needs further addressing in the study. The paper itself is written clearly and besides some minor issues is very suitable for publication in the journal.

Main comments

- The main issue with the technique applied to tropical Africa is related to the afore

mentioned high potential rates of soil erosion, which can cut deep into the saprolite through gully and badland formation. If soil is eroded from deeper saprolites (low activity) upslope and deposited downslope on surface soils, it lowers the activity of the plow layer because of sedimentation and not because of erosion. If I understand the model correctly, the main assumption is that lower activities will indicate erosion and higher rates sedimentation. How did you correct for the potential lowering of activities through sedimentation of erosion subsurface soils? I read in line 310 of the discussion that you use higher activities in the subsoil as a proxy for sedimentation, but there is not much mention of how this is incorporated in the mass balance model and estimations of soil redistribution. I think this effect (and the assumptions of the model) should be explained more clearly in the methodology. Moreover, these assumptions should also be included in the discussion as a limitation of the technique in the specific geographic context.

- Related to this comment is the potential effect of terracing. Were there terraces in the study fields? If so, I expect it would greatly influence the results of the model since you would get erosion and sedimentation patterns on a very small scale that might not be picked up by the sampling resolution.

- Why do you either use a mean forest references or a mean cropland plateau as reference. Wouldn't it be more relevant to pick a reference of a forested plateau soil closest to the cropland area?

Minor comments

Abstract: - Lines 18-19: After reading the introduction I know what you mean with the following statement 'challenging local conditions for long-term landscape scale monitoring'. However, when first reading the abstract I did not, so it might be beneficial to clarify this statement.

- Line 26: The most vulnerable regions of what? Soil erosion or socio-economically?

Introduction: - In my opinion, it would be beneficial to add a brief statement on the effects of soil erosion on sediment yields (see the review of Vanmaercke et al. 2014 about sediment yields in sub-saharan Africa) in the introduction. In that context, I'm also missing a mention of the climatic drivers of the high rates of soil erosion in the region (such as high intensity rainfall, intra- and interannual variations) in the introduction (you mention it in the method section).

- The discussion of the issues of population pressure, soil erosion and food security is portrayed quite linear (e.g. more people is more deforestation hence more soil erosion). There is evidence from Eastern Africa that the reality is not so linear (see: Tiffen 1994- More People, Less Erosion, or Wynants et al. 2019- Drivers of increased soil erosion in East Africa's agro-pastoral landscapes).

- Line 39: Social impact should be social impacts (plural).

Methods:

- Line 100: is there terracing or other soil conservation measures? Were the terraces slow-formed through natural processes or were they built? These deserve a mention since they are very influential for the redistribution of soil. Especially terracing can completely alter the 240Pu profile in a very short period. Figure 1 shows large scale vegetation patterns, that are not always relevant on the plot scale. In this context, it seems beneficial to the study to add some photos of the study areas (could also be as supplementary information).

- I would expand section 2.5. At the moment, you don't explain why you test these different scenarios and their statistical relevance.

Results:

- Do you have figures of soil redistribution for the forest sites?

Discussion:

- Could there be an effect of vegetation cover on the original fallout, wherein dense forest cover could influence the activities of fallout radionuclides in the soil? What was the vegetation cover situation during the time of peak fallout? These questions deserve some discussion.

- Line 353: when referencing values observed globally it is important to reference empirical studies and not studies where soil erosion values are modelled (such as the Borelli et al. 2017 study).

---

## Author Comment (AC2) · 15 Apr 2021

**Point-by-point response Referee #2**

**Assessing soil redistribution at forest and cropland sites in wet tropical Africa using $^{239+240}$Pu fallout radionuclides**

Florian Wilken, Peter Fiener, Michael Ketterer, Katrin Meusburger, Daniel Iragi Muhindo, Kristof van Oost, Sebastian Doetterl

We appreciate the reviewer sees the relevance of the study and supports publication. The reviewer calls for methodological clarification and discussion regarding the burial of eroded and already depleted sediments and the use of both cropland plateau
and forest sites as reference. We thank the reviewer for the valuable advices and revised the manuscript accordingly. Please see our detailed answers (in italics) to the comments below:

**Major reviewer comments**

The main issue with the technique applied to tropical Africa is related to the aforementioned high potential rates
of soil erosion, which can cut deep into the saprolite through gully and badland formation. If soil is eroded from deeper saprolites (low activity) upslope and deposited downslope on surface soils, it lowers the activity of the plow layer because of sedimentation and not because of erosion. If I understand the model correctly, the main assumption is that lower activities will indicate erosion and higher rates sedimentation. How did you correct for the potential lowering of activities through sedimentation of erosion subsurface soils? I read in line 310 of the discussion that
you use higher activities in the subsoil as a proxy for sedimentation, but there is not much mention of how this is incorporated in the mass balance model and estimations of soil redistribution. I think this effect (and the assumptions of the model) should be explained more clearly in the methodology. Moreover, these assumptions should also be included in the discussion as a limitation of the technique in the specific geographic context.

*We agree that source area depletion is a limitation of the mass balance model for soil redistribution assessments*
*in areas facing high soil loss rates. A way to understand the potential effect of $^{239+240}$Pu source depletion and corresponding sedimentation is to reduce the increment depth and thereby increase the number of samples to achieve more depth explicit $^{239+240}$Pu information. This information would be interesting, but increases the sample amount and associated costs substantially. With the sampling design applied in this study, which is based on two*

*layers from 0-60 cm and 60-100 cm at the cropland foot-slope sites, a very high topsoil depletion would be indicated by lower topsoil $^{239+240}$Pu activities compared to the deeper soil layer, which was not observed. Hence, our approach takes this process to some extent into account, but not on high resolution due to limitations in the amount of samples that were able to be analysed. The aim of the study is to understand the potential application of fallout*

*radio nuclides in tropical Africa. Therefore, we also sampled organic layers to understand $^{239+240}$Pu plant uptake and depth increments at plateau sites to understand a rather undisturbed depth distribution. For a follow-up study in tropical Africa, we would not analyse the organic layers in the forest but take an additional plough layer sample increment at foot-slope locations into account. In the revised version of the manuscript, we discuss the limitations of the mass-balance model and include a section for a suggested sampling design:*

*"The measured 239+240Pu activity at the foot-slope positions may underestimate the 239+240Pu inventory due to two potential processes: (i) Sedimentation exceeds the sampling depths that does not include the full inventory and (ii) deposition of sediments that are already depleted in 239+240Pu as the inventory at the eroded source area has been subject to pronounced soil and corresponding 239+240Pu loss. However, an indication for both processes would be an increasing 239+240Pu activity with depth, which is not reflected in the data (Fig. 3)."*

*"For future 239+240Pu based soil redistribution investigations in tropical cropland areas a two layered sampling scheme is suggested that individually samples the plough layer and the deeper soil layer (e.g. 0-20 cm and 20-60 cm at slope and an additional 60-100 cm sample at foot slope sites). The two layered sampling scheme accounts for the large number of samples falling below the detection limit and allows for a better understanding of the 239+240Pu depth distribution and thereby depletion status of the sediment source area. In tropical forests, the*

*sampling scheme can be focused on mineral soil as the activity in the litter and organic layer does not substantially contribute to the 239+240Pu inventory. Sedimentation at the three forest study sites did not exceed 60 cm at the foot slope, which suggests that a sampling depth of 0-60 cm is sufficient to include the full inventory. Nevertheless, additional deeper soil sampling, with reduced increment depth (e.g. 20 cm) at foot slope locations, is suggested to account for different environmental conditions and potentially higher foot slope sedimentation."*

Related to this comment is the potential effect of terracing. Were there terraces in the study fields? If so, I expect it would greatly influence the results of the model since you would get erosion and sedimentation patterns on a very small scale that might not be picked up by the sampling resolution.

*We fully agree that terraces lead to very complex erosion and deposition processes that would be reflected in the*
*$^{239+240}$Pu inventories. However, the study sites do not show any terracing, which is in the DR Congo and Ugandan*

*parts not common. This is not true for Rwanda, which where terracing is common for cropland slopes. To make
this clear, we point at non-terracing in the study area description as follows:*

*"The cropland sites represent the typical smallholder farming found in the region, which is based on small non-
terraced fields with non-mechanised tillage practices."*

Why do you either use a mean forest reference or a mean cropland plateau as reference. Wouldn't it be more
relevant to pick a reference of a forested plateau soil closest to the cropland area?

*We found a large discrepancy between the plateau sites of forests and croplands. In theory, both positions should
be rather similar, as they have not been subject to substantial soil loss or sedimentation by tillage and water and*

*received more or less the same fallout. Solely using the plateau sites would ignore potential losses of $^{239+240}Pu$ due
to cropland use (e.g. harvest erosion of root crops), which might cause an altered reference. As we cannot name
the correct reference, considering both as potential references is important to demonstrate uncertainties to the
reader. To clarify this, we add the following information in the discussions.*

*"In this study, two different reference scenarios are taken into account to address potential differences of $^{239+240}Pu$*

*inventories between stable forest and cropland plateau positions: (i) $Ref_{for}$ mean of specific forest sites (ii) and
$Ref_{plt}$ mean of cropland plateau positions of the specific sites."*

Minor comment:

Abstract:

Lines 18-19: After reading the introduction I know what you mean with the following statement 'challenging local
conditions for long-term landscape scale monitoring'. However, when first reading the abstract I did not, so it might
be beneficial to clarify this statement.

*Thanks! We modify the wording by adding "infrastructure limitations":*

*"However, there is limited knowledge on soil redistribution dynamics following land conversion into arable land*

*in tropical Africa that is partly caused by infrastructure limitations for long-term landscape scale monitoring."*

Line 26: The most vulnerable regions of what? Soil erosion or socio-economically?

*Thank you for this hint. We change the wording as follows:*

*"[…] in one of the most socio-economically and ecologically vulnerable regions of the world."*

Introduction:

In my opinion, it would be beneficial to add a brief statement on the effects of soil erosion on sediment yields (see the review of Vanmaercke et al. 2014 about sediment yields in sub-saharan Africa) in the introduction.

*To point out that soil erosion does not only lead to soil redistribution but to huge amounts of sediment delivery (impressively shown in the high sediment load of rivers), we explicitly name sediment delivery as an important process affecting the region as follows:*

*"In particular, the White Nile-Congo rift (NiCo) region faces a strong impact of soil redistribution (Lewis and Nyamulinda, 1996; FAO and ITPS, 2015; Montanarella et al., 2016) and corresponding sediment delivery (Vanmaercke et al., 2014) due to steep terrain, high rainfall erosivity with a strong intra-annual seasonality (Fick and Hijmans, 2017) that causes sparse vegetation cover of the soil at the end of the dry seasons but also throughout the cultivation period (Lewis and Nyamulinda, 1996) due to non-mechanised farming."*

In that context, I'm also missing a mention of the climatic drivers of the high rates of soil erosion in the region (such as high intensity rainfall, intra- and interannual variations) in the introduction (you mention it in the method section).

*We follow the suggestion and add statements on the environmental drivers in the Introduction:*

*"In particular, the White Nile-Congo rift (NiCo) region faces a strong impact of soil erosion (Lewis and Nyamulinda, 1996; FAO and ITPS, 2015; Montanarella et al., 2016) due to steep terrain, high rainfall erosivity with a strong intra-annual seasonality (Fick and Hijmans, 2017) that causes sparse vegetation cover of the soil at the end of the dry seasons but also throughout the cultivation period (Lewis and Nyamulinda, 1996) due to non-mechanised farming."*

The discussion of the issues of population pressure, soil erosion and food security is portrayed quite linear (e.g. more people is more deforestation hence more soil erosion). There is evidence from Eastern Africa that the reality is not so linear (see: Tiffen 1994- More People, Less Erosion, or Wynants et al. 2019- Drivers of increased soil erosion in East Africa's agro-pastoral landscapes).

*We agree that population pressure causes complex processes that do not lead to a linear increase of soil erosion. Our main intention is to point out that an increasing food demand is typically compensated by deforestation, which is the prerequisite for substantial soil erosion. After land use conversion, sustainable and unsustainable agricultural use might be applied, which is –indeed- not linear. We add "and often unsustainable use of soil*

*resources" to the sentence to clarify that unsustainable use of soil resources is 'often' practiced but not necessarily the case:*

*"[…] the onset of potential soil erosion at previously undisturbed sites (Nyssen et al., 2004) and often unsustainable use of soil resources (Wynants et al., 2019)."*

Line 39: Social impact should be social impacts (plural).

*Thanks, done!*

Methods:

Line 100: is there terracing or other soil conservation measures? Were the terraces slow-formed through natural processes or were they built? These deserve a mention since they are very influential for the redistribution of soil. Especially terracing can completely alter the 240Pu profile in a very short period.

*We agree, but there is no terracing in both cropland study sites. To clarify this, we add this information to the test-site description:*

*"[…] based on small non-terraced fields with non-mechanised tillage practices."*

Figure 1 shows large scale vegetation patterns, that are not always relevant on the plot scale. In this context, it seems beneficial to the study to add some photos of the study areas (could also be as supplementary information).

*We add orthophotos and a panorama to the supplements to illustrate the landscape.*

*Figure S1*

I would expand section 2.5. At the moment, you don't explain why you test these different scenarios and their statistical relevance.

*Thanks, we add explanation and the reasons behind the different scenarios in section 2.5 as follows:*

*"The mass balance soil mixing model was used to assess different scenario assumptions and their sensitivity. First, different 239+240Pu reference inventories were determined in two ways: (i) the mean 239+240Pu inventory of all forest sites of a specific region (Reffor; i.e. mean inventory of Kahuzi Biega forest for the DR Congo cropland sites and Kibale forest for the Ugandan cropland sites) and (ii) the mean 239+240Pu inventory of the cropland plateau sites of the specific region (Refplt). These land use specific reference scenarios are supposed to address potential*

*differences between Reffor and Refplt to understand uncertainties associated with the reference determination. Second, the sensitivity of the ploughing and corresponding mixing depth is assessed using a 20±5 cm ploughing*

*depth deviation as the exact plough depth over the integration period cannot be accurately derived. Third, to address potential interannual variability of water erosion, a scenario with five extreme years producing the same total soil erosion as a 55 years continuous soil erosion rate was compared against the results of the first scenario."*

Results:

Do you have figures of soil redistribution for the forest sites?

*As our results show that no substantial soil erosion is taking place at the forest sites, we are not showing a forest soil redistribution Figure. We add a sentence in the Discussion to make it more clear that soil redistribution at the forest sites does not exceed the effect of natural variability:*

*"The variation of forest 239+240Pu inventories due to bioturbation and fallout infiltration patterns (e.g. caused by throughfall or stem flow patterns) exceeds a potential soil redistribution impact [...]. Hence, the forest sites are assumed to represent almost the entire 239+240Pu inventory of the baseline inventory without substantial loss due to soil erosion."*

Discussion:

Could there be an effect of vegetation cover on the original fallout, wherein dense forest cover could influence the activities of fallout radionuclides in the soil? What was the vegetation cover situation during the time of peak fallout? These questions deserve some discussion.

*The cropland and cropland reference site in DR Congo was arable land at the time of nuclear weapon tests, while*
*the study site in Uganda was converted into cropland after the nuclear weapon tests (approx. 1970s). The effect of forest cover on the mean 239+240Pu inventory is expected to be low. However, as higher fallout heterogeneities are typically found in forests (stem flow, canopy through fall patterns etc.; see Discussions section "Soil redistribution in cropland of the NiCo region"), spatial $^{239+240}$Pu variability in the Ugandan study site might be higher. However, these initial patterns are likely to be homogenised to a certain extent due to lateral soil mixing*
*by ploughing. We expand the discussion on the different land use conditions in the cropland test sites as follows:*
*"The different land use during the main time of atmospheric nuclear weapon tests (DR Congo=cropland; Uganda=forest) might have caused higher spatial variability of the initial 239+240Pu inventory associated with complex infiltration (e.g. stem flow) and throughfall patterns in forests (Hofhansl et al., 2012). However, these small-scale variation is likely to be homogenised after 40 years of lateral soil movement by plough-based arable use."*

Line 353: when referencing values observed globally it is important to reference empirical studies and not studies where soil erosion values are modelled (such as the Borelli et al. 2017 study).

*Thanks for pointing at this. We change the text as follows:*

*"In comparison to values observed (Boardman and Poesen, 2006) and modelled globally (Borrelli et al., 2017)*

*[...]"*

---

## Author Comment (AC1)

**Point-by-point response Referee #1**

**Assessing soil redistribution at forest and cropland sites in wet tropical Africa using $^{239+240}$Pu fallout radionuclides**

5 Florian Wilken, Peter Fiener, Michael Ketterer, Katrin Meusburger, Daniel Iragi Muhindo, Kristof van Oost, Sebastian Doetterl

We are pleased about the reviewer acknowledging the study and supports publication. We highly appreciate the valuable advices and revised the manuscript accordingly. Please see our detailed answers (in italics) to the comments below:

**Major reviewer comments**

In the Methods, adding a table comparing the main characteristics of the 5 investigated sites would be very helpful for the readers. For instance, text on page 4 is hard to follow and an additional table would definitely help comparing the site characteristics at a glance.

15 *Thank you for this idea. We will provide a table showing the following information: (i) latitude and longitude, (ii) mean slope of the test sites, (iii) mean annual rainfall according to Fick and Hijmans 2017, (vi) mean annual air temperature and (v) mean field size.*

On L.116, you refer for the 1st time to "subsoil" (which then appears on many occasions in the text). In my opinion, this is

20 misleading as you are referring to footslope locations (you even refer to "colluvial sites" in the text) so that – in my opinion – you expect to find deep accumulations of eroded soil at those locations which likely mainly consist of "eroded topsoil" which may be progressively buried at the footslope » you should rather refer to "deeper soil layers" and avoid using the "subsoil" term … This should be corrected throughout the entire manuscript.

*Thanks, we removed the term subsoil and in most cases used the term "deeper soil layer".*

25

Regarding the calculation of soil redistribution rates, I wondered when reading section 2.4 how you took the reference inventories into account, then when reading section 2.5, I likely found the answer: you use it as an important factor in the sensitivity analysis given the large potential uncertainties, right? Do you confirm this?

*Yes, we include the reference inventory in the uncertainty analysis. To make this already in section 2.4 clear, we changed the equation caption as follows:*

*"where R is the soil redistribution rate in m (55 yrs)$^{-1}$, $A_{ref}$.is the $^{239+240}$Pu reference inventory and A is the $^{239+240}$Pu activity difference (reference vs. local activity) in m x m (55 yrs)$^{-1}$ and Bq kg$^{-1}$ x Bq kg$^{-1}$ (55 yrs)$^{-1}$ over the simulation period (more*

5   *details on the $A_{ref}$ implementation are given in section 2.5)."*

*Furthermore, we elaborate on the reasons for the different scenarios in section 2.5 as follows:*

*"The mass balance soil mixing model was used to assess different scenario assumptions and their sensitivity. First, different $^{239+240}$Pu reference inventories were determined in two ways: (i) the mean $^{239+240}$Pu inventory of all forest sites of a specific region (Ref$_{for}$; i.e. mean inventory of Kahuzi Biega forest for the DR Congo cropland sites and Kibale forest for the Ugandan*

10   *cropland sites) and (ii) the mean $^{239+240}$Pu inventory of the cropland plateau sites of the specific region (Ref$_{plt}$). These land use specific reference scenarios are supposed to address potential differences between Ref$_{for}$ and Ref$_{plt}$ to understand uncertainties associated with the reference determination. Second, the sensitivity of the ploughing and corresponding mixing depth is assessed using a 20±5 cm ploughing depth deviation as the exact plough depth over the integration period cannot be accurately derived. Third, to address potential interannual variability of water erosion, a scenario with five extreme years producing the*

15   *same total soil erosion as a 55 years continuous soil erosion rate was compared against the results of the first scenario."*

In the discussion, on pages 10-11, in my opinion, there is one hypothesis lacking to explain the low inventories measured in this study. What about the potential export of soil/plutonium from the hillslope to locations located further downstream? Are the hillslopes connected to lower locations, how is the connectivity between the hillslopes and lower zones? We may imagine

20   that part of the eroded soil and the associated plutonium have been exported from the hillslopes during the last several decades.

*Thanks, we fully agree that deposition of low $^{239+240}$Pu activity soil is a limitation of the approach and could hide sedimentation at the foot slope positions. However, in an extreme case, an increasing $^{239+240}$Pu activity with depth would be expected, which is not the case (Fig. 2). Furthermore, the foot-slope $^{239+240}$Pu inventory at the cropland study site in Uganda does not exceed the corresponding plateau inventory, which would have been the case if substantial sedimentation has taken place.*

25   *We expand the discussion on this topic as follows:*

*"The measured 239+240Pu activity at the foot-slope positions may underestimate the 239+240Pu inventory due to three potential processes: (i) Sedimentation exceeds the sampling depths that does not include the full inventory and (ii) deposition of sediments that are already low in 239+240Pu activity at the eroded source soil has been subject to pronounced soil and corresponding 239+240Pu loss and (iii) sedimentations sites may temporarily turn into erosion sites. However, an indication*

30   *for these processes would be an increasing 239+240Pu activity with depth, which is not reflected in the data (Fig. 3)."*

Furthermore, on LL.272-280, maybe in addition to rainfall depth, it is important to take into account the latitude of the investigated sites, as fallout radionuclide inventories were showed to be strongly latitude-dependent (the location in the Northern or the Southern Hemisphere is also of importance).

*We add more information on the relationship between latitude and 239+240Pu inventories as follows:*

*"It is well established that the initial deposition of 239+240Pu fallout was strongly dependent upon latitude; the study of Kelley et al. (1999) compared 237Np, 239Pu and 240Pu inventories from soil cores collected from undisturbed, worldwide locations. One data point (Muguga, Kenya) included in the results of Kelley et al. (1999) yields a 239+240Pu inventory of*

5   *19.2 Bq m-2, which is significantly lower than the inventories for soils in the mid-latitude regions of the Northern Hemisphere, e.g., Munich, Germany exhibits a 239+240Pu inventory of 86 Bq m-2. Although the results of Kelley et al. (1999) are drawn from a limited set of samples, they nevertheless make the case for a strong dependency of inventory vs. latitude, as well as annual precipitation, as factors that control the depositional inventory."*

*Furthermore, to illustrate the direct proximity of the sites to the equator (max distance to equator 2.5°), we add the latitude*

10   *and longitude coordinates in the suggested test site table.*

Finally, in the discussion, maybe adding a sequence on recommendations for future studies using fallout radionuclides in tropical environments would be useful. For instance, the results described on LL.205-209 confirm that, in future, the analysis of L/O horizon samples could be avoided because they contain only negligible proportions of the total plutonium inventories

15   (this would save time to analyse more soil depth increments for instance).

*Thanks, this is a good idea. We include a short section on recommendations for future $^{239+240}$Pu based soil redistribution investigations in the tropics:*

*"For future 239+240Pu based soil redistribution investigations in tropical cropland areas, a two-layered sampling scheme is suggested that individually samples the plough layer and the deeper soil layer (e.g. 0-20 cm and 20-60 cm at a slope and*

20   *an additional 60-100 cm sample at foot slope sites). The two-layered sampling scheme accounts for the large number of samples falling below the detection limit and allows for a better understanding of the 239+240Pu depth distribution and thereby depletion status of the sediment source area. In tropical forests, the sampling scheme can be focused on mineral soil as the activity in the litter and organic layer does not substantially contribute to the 239+240Pu inventory. Sedimentation at the three forest study sites did not exceed 60 cm at the foot slope, suggesting that a sampling depth of 0-60 cm is sufficient to*

25   *include the full inventory. Nevertheless, additional deeper soil sampling, with reduced increment depth (e.g. 20 cm) at foot slope locations, is suggested to account for different environmental conditions and potentially higher foot slope sedimentation. The determination of appropriate reference sites is critical. At plateau sites, three depth increments (0-20 cm; 20-40 cm; 40-60 cm) were sampled to understand the depth distribution of sites without substantial water erosion. At the plateau sites, no 239+240Pu activity was found in the 40-60 cm soil layer. Hence, a single sample that integrates the depth*

30   *from 0 to 40 cm seems to be sufficient. However, the number of reference samples should be high enough to represent a robust mean, particularly at reference sites that are under arable use."*

**Detailed remarks Title**

Maybe the title could be slightly improved, for instance: "assessing soil redistribution at forest and cropland sites …" (as you quantified soil erosion and accumulation?)

*Thanks, we follow your recommendation.*

**Abstract**

L.18 (and all throughout the manuscript, I didn't list all the word occurrences): conversion INto arable land (instead of in)

*Many thanks, done.*

10   L.18 unclear what you mean with "challenging local conditions"

*Thanks, we changed the text to "infrastructure limitations".*

L.21 "a relatively high inventory" » as this is all relative, maybe you could adapt the phrasing; e.g. this is definitely not high compared to the inventories observed in other regions of the world.

15   *We agree and indicate in the text that the inventory is relatively high for a tropical study area:*

*"In the study area, a $^{239+240}Pu$ baseline inventory is found that is higher than typically expected for tropical regions (mean forest inventory 41 Bq m$^{-2}$)."*

L.23 "up to 37 and 40 cm" » maybe provide the soil redistribution rates instead?

20   *We added the requested information as follows:*

*"[...]soil erosion and sedimentation on cropland reached up to 37 cm (81 Mg ha$^{-1}$ yr$^{-1}$) and 40 cm (87 Mg ha$^{-1}$ yr$^{-1}$) within the last 55 years, respectively."*

L.25 "insight into" instead of "insight on"

25   *Thanks, done!*

L.26 Does world deserve a capital letter here?

*Thanks, you are right, it does not! We changed it.*

30   **Introduction**

L.33 "low soil cover conditions" » sparse vegetation cover of the soil?

*Thanks, we followed your recommendation.*

L.35 maybe specify that you refer to 'CROP yield'

*Thanks, we followed your recommendation to use the more specific term "crop yield"*

L.35 'goes hand in hand' » maybe consider rephrasing?

*We changed the text as follows:*

5   *"The loss of soil resources and crop yield decline in the NiCo region happens parallel with a rapid population growth [...]"*

L.37 for THE entire Sub-Saharan Africa?

*Thanks, done!*

10   LL.40-41 "to assess new non-degraded soils" » unclear, please rephrase

*We changed the text as follows:*

*"Under current practices, an increasing demand in food is typically compensated through deforestation to assess new soil resources, which are often located in areas with steep slopes."*

15   L.42-43: "the onset of soil erosion", maybe specify "at previously unaffected/pristine sites"?

*We changed the wording as follows:*

*"[...]onset of soil erosion at previously undisturbed sites."*

L.45 "loss of potential reforestation" » impossibility of reforestation?

20   *We changed the text as follows:*

*"[...], which means a quasi-permanent loss of cropland and the option to reforest areas for decadal to centennial timescales."*

L.48 at suitable locations

25   *Thanks, we follow your suggestion?*

L.50 is detailed information > requires detailed information?

*We changed the text as follows:*

*"To develop a smart intensification plan detailed information on soil degradation dynamics of individual regions under*

30   *specific conditions is essential (e.g. land use, topography, soil type, rainfall characteristics)."*

L.50 please avoid the repetitive use of 'specific'

*Thanks, please see the response above.*

L.54 "but are important" » although they remain crucial?

*Thanks, we follow your suggestion.*

L.55 insight of > into

*We change the text as follows: "[…]that provides insight into […]"*

L.55 "internal soil redistribution dynamics" » do you mean at the hillslope scale?

*We changed the text as follows: "[…]that provides insight into spatially distributed soil redistribution dynamics[…]"*

L.58 "overcome by fallout radionuclides" » by the analysis of fallout radionuclides?

*We follow your suggestion.*

L.60 on > into

*Thanks, done.*

L.64 of the 137Cs activity until today?

*Thanks for pointing at this!*

L.64 in tropical and equatorial regions?

*We follow your suggestion.*

L.65 to much lower fallout

*We removed "a".*

LL.66-67 extreme erosion rates… » there seems to be a verb missing in this sentence

*Thanks, we change the text as follows:*

*"Furthermore, extreme soil erosion rates in the tropics are additionally driving a depletion of the $^{137}$Cs inventories."*

L.67 Over the past decade… » there is a transition missing from the previous sentence here, in my opinion. Maybe add "To overcome these analytical difficulties…"?

*Thanks, we changed the sentence as follows:*

*"To overcome these analytical difficulties, the fallout radionuclides $^{239}$Pu and $^{240}$Pu have been discussed and tested as an alternative radioisotopic tracer to $^{137}$Cs for soil redistribution studies over the past decade."*

L.69 their long half-life?

*Thanks, we follow your suggestion.*

L.69 "without relevant decay" » this is not needed and should be removed

5 *We follow your suggestion.*

L.73 provide important insights into…

*Thanks, we follow your suggestion.*

10 L.74 "where, to our best knowledge,…" » and, to the best of our knowledge, none was conducted…

*Thanks, we follow your suggestion.*

LL.76-78 this should be rephrased in my opinion (strange to start with "we follow two major aims"; "exemplarily analysing" does not read well…)

15 *We rephrased the text as follows:*

*"The aims of the study are (i) to test the suitability of fallout radionuclides $^{239}$Pu and $^{240}$Pu as tracers of soil redistribution in the wet Tropics of Africa, and (ii) to analyse soil redistribution dynamics after conversion from forest into cropland for selected sites within the East African NiCo region."*

20 **Methods**

L.81 Lake Eduard » Edward?

*Thanks, done!*

L.83 conversion of forest into cropland?

25 *Thanks, we follow your suggestion.*

L.85 gully > gullies

*Thanks, done!*

30 L.86 sum rainfall? I would remove 'sum'

*Thanks, we follow your recommendation.*

L.87 maybe add the range of years during which this precipitation was measured?

*We add the time range and reference:*

*"(time period 1970-2000; Fick and Hijmans, 2017)"*

L.87 subdivided into two cycles of wet and dry seasons?
*Thanks, we follow your suggestion.*

L.88 storm events with large rainfall amounts?
*Thanks, done!*

L.92 throughout > across?

10 *Thanks, we follow your suggestion.*

L.96-97 1950s / 1970s
*Thank you for pointing at, done!*

15 L.99 "soil cover conditions are very patchy" » please rephrase
*We changed the text as follows:*
*"[...] a high patchiness of soil cover conditions is present and can alternate between bare soil and fully grown vegetation cover in direct proximity to each other."*

20 L.100 "in direct proximity" » not sure how Fig. 1 really illustrates this issue as I guess that the pixel size on Fig. 1 is not compatible with intra-field heterogeneities within the study sites?
*We fully agree and remove the reference to the Figure.*

L.110 "in order to understand variation of radionuclide inventories at sites" » do you mean "in order to quantify spatial
25 variations in reference fallout inventories"?
*Thanks, we follow your suggestion.*

L.112 L and O horizons
*Thanks, done!*

30

L.122 "from the study slope" » unclear what you mean here
*We changed the word study site into study slope to avoid this misleading formulation:*

*"In DR Congo, a cropland plateau site (converted to grassland approximately in 2005) located about 8 km apart from the study site was sampled, while in Uganda, flat plateau sites under arable use in direct proximity to the study site were sampled."*

5    L.171 "different nature" » unclear what you mean here

*We reformulated the sentence as follows:*

*"Soil erosion and sedimentation processes were implemented (R-Core-Team, 2019) individually to account for differences of both processes."*

10    L.175 processes

*Thanks, done!*

L.191 from the literature, i.e. 80 kg m-3

*Thanks, done!*

Results (please avoid the use of the term "subsoil" in this section)

*We consistently removed the term subsoil from the manuscript.*

L.204 it seems that Fig. 3 is cited in the text before Fig. 2, maybe their sequence should be reversed.

20    *Thanks for pointing us to this error. We swapped the sequence of Fig. 2 and Fig. 3.*

L.208 rarely » maybe indicate the % instead?

*We changed the text as follows:*

*"Few (13%, 7 of 55 samples) of $^{239+240}$Pu activities of the 0 to 60 cm depth mineral soil layers fall below the detection*

25    *limit[…]"*

L.210 Why do you consider subsoil as the layer lower as 60 cm?

*We removed the term and consistently use "deeper soil layer" instead.*

30    L.214 within each forest site?

*Thanks, done!*

L.216 'that falls in range by two standard deviations of the plateau mean' » unclear what you mean here, please rephrase

*We rephrased the text and removed the reference to Figure 3 to avoid confusion between activity and inventories, which are the focus in this sentence.*

*"The mean $^{239+240}$Pu inventories found at the slope and foot-slope sites fall within the range of one standard deviation ± mean $^{239+240}$Pu inventory of the corresponding plateau sites."*

LL.219-224: I have the impression that part of the text here is repeated from the previous paragraphs, please check and avoid repetitions

*The two sections are somewhat linked as they are supposed to show the difference between the forest (section: $^{239+240}$Pu activities and inventories of forest sites) and cropland (section: $^{239+240}$Pu activities and inventories of cropland sites) sites, which might let them appear similar. We checked for repeated wording and highlight the difference between the two sections by rephrasing the section heading to*

*"Forest sites $^{239+240}$Pu activities and inventories" and "Cropland sites $^{239+240}$Pu activities and inventories".*

L.221 "0.019_0.006 Bq kg-1." » can we really be confident with 3 decimal digit significance here?

*We agree and replace the 0.006 by <0.01 Bq kg$^{-1}$: "a mean and standard deviation of 0.019±<0.01 Bq kg$^{-1}$".*

L.224 at slope > upslope?

*We would like to stay with the term "slope" instead of "upslope". From our perspective, using the term "upslope" could be misleading and excludes the mid and downslope section.*

L.233 "similar" » similarly?

*Thanks, done!*

L.244 and in contrast … using … » please rephrase

*We rephrased the sentence as follows:*

*"[…], while using Ref$_{plt}$ at least 10 cm (55 yrs)$^{-1}$ of soil loss must have taken place before the detection limit is reached."*

L.249 "testing the concentrated scenario" » this is probably not the best wording (extreme rainfall scenario?)

*We agree that the term is not well-fitting. We changed the wording to a somewhat bulky but precise scenario description:*

*"The low frequency but high magnitude soil erosion scenario […]"*

L.250 "after 19% less total soil loss"» confusing, please rephrase

*We rephrased the sentence as follows:*

*"[…] showed that detection limit is reached already at 81% of soil loss compared to the continuous 55 years erosion rate."*

L.253 "widely" > strongly?

*We follow your suggestion.*

5   L.256 "sloping positions" » unclear

*We remove the term sloping and consistently use the term "slope" sites throughout the manuscript.*

L.256 weaker > lower?

*Thanks, we follow your recommendation.*

L.261 'subsoil" » deep soil layers?

*We consistently follow your recommendation throughout the manuscript.*

L.265 selective transport: are you referring to particle size here?

15   *Yes, to make this clearer, we add the following to the sentence:*

*"[…]enrichment processes by selective transport of fine soil particles."*

**Discussion**

L.268 "within this study" > in this study?

20   *Thanks, done!*

L.272 "inventory findings" » inventory is found?

*Thanks, we changed the wording to:*

*"The $^{239+240}$Pu inventories found […]".*

25

L.280 high for conducting soil redistribution studies

*Thanks, done!*

L.287 to cover » to include?

30   *Thanks, done!*

L.290 "within this study" > in this study?

*Thanks, we follow your correction!*

L.296 small > low?

*Thanks, we follow your suggestion.*

L.305 "represent almost the entire 239+240Pu inventory of the global fallout" » are you referring to the reference/baseline inventories here?

*We change the text as follows:*

*"[...]of the baseline inventory."*

L.310 corresponding fallout patterns » are you referring to their heterogeneities in particular here?

*We refer here to spatial patterns of the baseline inventory, which can be driven by the spatial distribution of rainfall. To make this clearer, we change the text as follows:*

*"[...]which cannot be explained by local rainfall heterogeneities and corresponding spatial patterns of the baseline inventory."*

L.313 at play not investigated by this study » that were not investigated by this study

*Thanks, done!*

L.315 "in subsoil" > with depth

*Thanks, done!*

L.317 like that observed for the fallout: : :

*Thanks, done!*

L.318 activity in crops » it is particularly very unlikely to find high Pu activities in vegetation 60 years after the fallout…

*The intention of this sentence was to point out that $^{239+240}$Pu plant uptake is limited and not a reasonable explanation for the substantially lower inventories. We slightly rephrased the text to make the statement clearer:*

*"[...]is unlikely as no elevated $^{239+240}$Pu activity was found in harvested crops of other studies (Akleyev et al., 2000)."*

L.330 on the contribution » of the contribution?

*Thanks, done!*

L.337 falling below[…] » with activities falling below[…]

*Thanks, done!*

L.340 "varying length" » do you mean "duration" here? Or the period since the conversion of tropical forest into cropland?

*Thanks, we rephrased the text as follows:*

*"We relate this discrepancy of the different duration since DR Congo and Uganda tropical forest has been converted into cropland."*

L.342 "55 years" » shouldn't this 55-yr period be adapted depending on the duration since the conversion of forest into cropland?

*We agree and refer the soil loss rates to 40 yrs. as follows:*

*"[…] Uganda (referred to 40 yrs. of arable land use and corresponding soil redistribution): -18.4 Mg ha$^{-1}$ yr$^{-1}$ Ref$_{for}$, 27.8 Mg ha$^{-1}$ yr$^{-1}$ Ref$_{plt}$ […]"*

L.345 "cropland use" » cultivation period?

*We use the term:*

*"cropland cultivation period".*

L.347 were » was?

*Thanks, done!*

L.349; what about the occurrence of crop rotations in the different zones of interest?

*This sentence's main purpose is to point out differences in the main cultivated plant species that might have an impact on soil cover and harvest erosion processes. To clarify this, we reformulated the sentence as follows:*

*"However, the main cultivated crops differ between the Ugandan and DR Congo study sites substantially. In Uganda, sweet potato and maize are the major crops, while DR Congo cropland is dominated by cassava. This difference may have an impact on water and harvest driven soil erosion processes."*

L.354 "within the region" » observed in the region?

*Thanks, we follow your suggestion!*

L.355 "The range of observed values at slopes spans from net sedimentation to heavy soil loss in direct proximity to each other" » unclear, please rephrase

*We rephrased the text as follows:*

*"Over very short distances, high soil redistribution heterogeneities ranging from sedimentation to heavy soil loss are found, which might be an effect of smallholder farming structures. These farming structures are potentially mitigating […]"*

**Conclusions**

L.362 usability » feasibility of using/analysing?

*Thanks, we follow your recommendation as follows:*

*"[…]the feasibility of analysing […]"*

L.364 catena > catenae?

*Thanks, done!*

L.365 "indicative for little to no soil erosion" » which demonstrates the (almost) absence of erosion?

*We changed the text as follows:*

*"[…], which points at minor soil erosion."*

L.367 "However, the selection of an appropriate reference is critical due to a potential 239+240Pu inventory reduction by harvest erosion in root crop dominated cropland systems." » it seems to go pretty far in the interpretation here, focusing on the magnitude of root crop erosion, which has not been quantified in the current research;

*To avoid overinterpretation, we generalise the statement as follows:*

*"However, the selection of an appropriate reference is critical due to a potential 239+240Pu inventory reduction associated with cropland use other than water erosion."*

I wonder what would be the importance of sediment export from the hillslope (see comment above)

*Thanks we would like to refer to our answer given above.*

Figures (overall, your figures are beautiful, congratulations for that!)

*Thank you for this supporting comment and the appreciation!*

Fig. 1 maybe add the latitudes on this map (at least the Equator should be added); what is the source of the land use data?

*Including the Equator is an excellent idea that helps for orientation. We include the Equator in the Figure.*

Fig. 3, caption L.531 » illustrates

*Thanks, done!*

Fig. 4, caption L. 538: "limit of […] " » limit with different […]?

*Thanks, done!*

Fig.5, caption L. 546: were analysed » were used for calculation?

*Thanks, done!*

Tables

5    Table 1; it may seem counter-intuitive to analyse depth increments on the plateau and not at the footslope where sediment
accumulates?

*Yes, this is right. The reason behind this was to understand the depth profile and reference inventory at non-eroded sites to
evaluate the potential of $^{239+240}$Pu for soil redistribution assessments in the study region. In relation to this, we provide a
suggested sampling scheme for reference site sampling:*

10   *"The determination of appropriate reference sites is critical. At plateau sites, three depth increments (0-20 cm; 20-40 cm;
40-60 cm) were sampled to understand the depth distribution of sites without substantial water erosion. At the plateau sites,
no 239+240Pu activity was found in the 40-60 cm soil layer. Hence, a single sample that integrates the depth from 0 to 40
cm seems to be sufficient. However, the number of reference samples should be high enough to represent a robust mean with
regards to 239+240Pu variability typically associated with the corresponding land use."*

Caption L. 513: why not in Rwanda?

*In Rwanda, only forest sites were analysed due to a limitation of $^{239+240}$Pu samples to be analysed. We rather invested in
understanding the baseline inventory instead of another cropland site. The sentence might have stressed this too much. To
avoid this, we reformulate the text as follows:*

20   *"Numbers of samples taken at three forest (DR Congo, Uganda, Rwanda) and two cropland (DR Congo, Uganda) study sites.
While the O horizon depth is measured individually for each sample, the L horizon depth is assumed to be 1 cm as it was not
possible to be accurately measured. L = L horizon; O = O horizon; M = mineral layer 1: 0-60 cm and 2: 60-120 in forest and
60-100 cm in cropland."*